

# Influence of the North Atlantic Oscillation on European tropospheric composition: an observational and modelling study

**Richard J. Pope**[1,2]**, Martyn P. Chipperfield**[1,2]**, Stephen R. Arnold**[1]**, Norbert Glatthor**[3]**, Wuhu Feng**[1,4]**, Sandip S. Dhomse**[1]**, Brian J. Kerridge**[5]**, Barry G. Latter**[5]**, and Richard Siddans**[5]

[1]School of Earth and Environment, University of Leeds, Leeds, UK
[2]National Centre for Earth Observation, University of Leeds, Leeds, UK
[3]Karlsruhe Institute of Technology, Institute of Meteorology and Climate Research, Karlsruhe, Germany
[4]National Centre for Atmospheric Science, University of Leeds, Leeds, UK
[5]Remote Sensing Group, STFC Rutherford Appleton Laboratory, Harwell Oxford, UK

Correspondence to: Richard Pope
(r.j.pope@leeds.ac.uk)



## Abstract

We have used satellite observations and a simulation from the TOMCAT chemistry transport model (CTM) to investigate the influence of the well-known winter-time North Atlantic Oscillation (NAO) on European tropospheric composition. Under the positive phase of the NAO (NAO+), strong westerlies tend to enhance transport of European pollution (e.g. nitrogen oxides, $NO_x$, carbon monoxide, CO) away from anthropogenic source regions. In contrast, during the negative phase of the NAO (NAO-), more stable meteorological conditions lead to a build up of pollutants over these regions, relative to the winter-time average pollution levels. However, the secondary pollutant ozone shows the opposite signal of larger values during NAO+. NAO+ introduces Atlantic ozone-enriched air into Europe while under NAO- westerly transport of ozone is reduced yielding lower values over Europe. Furthermore, ozone concentrations are also decreased by chemical loss through the reaction with accumulated primary pollutants such as nitric oxide (NO) in NAO-. Peroxyacetyl nitrate (PAN), in the upper troposphere-lower stratosphere (UTLS) peaks over Iceland/Southern Greenland in NAO-, between 200-100 hPa, consistent with the trapping by an anticyclone at this altitude. Model simulations show enhanced PAN over Iceland/Southern Greenland in NAO- is associated with vertical transport of polluted air from the troposphere into the UTLS. Overall, this work shows that NAO circulation patterns are an important governing factor for European winter-time composition and air pollution.

## 1 Introduction

Atmospheric circulation can play an important role in the transport and accumulation of air pollutants from and over source regions (e.g. Pope et al. (2014); Stohl (2006); Quinn et al. (2007)). This is most evident in the Northern Hemisphere winter/spring when emission of anthropogenic pollutants (e.g. nitrogen oxides ($NO_x$) and carbon monoxide (CO)) are largest (Edwards et al., 2004; Zhou et al., 2012), slower chemical



loss mechanisms (i.e. photochemistry and reaction with OH) remove less pollution and winter-time dynamics are enhanced (more intense mid-latitude depressions and blocking systems (Trigo, 2006; Hurrell and Deser, 2010)). Over North America, the North Atlantic and Europe, the winter-time North Atlantic Oscillation (NAO) is one of the most prominent and frequent modes of atmospheric variability. It represents the redistribution of atmospheric mass between the Arctic and subtropical North Atlantic (Hurrell and Deser, 2010) controlling pressure gradients, wind flows, storm tracks and moisture budgets (Hurrell, 1995; Osborn, 2006). During the NAO positive phase (NAO+), the climatological Icelandic low and Azores high pressure systems both intensify leading to enhanced westerly circulation (storm tracks) across the Atlantic into north-western Europe. The NAO negative phase (NAO-) results in a weakening of this meridional pressure gradient leading to reduced westerly winds and a re-orientation of the storm tracks over southern continental Europe.

Previous studies have used a range of satellite observations and modelling tools to investigate the impact of NAO circulation patterns on tropospheric composition. Eckhardt et al. (2003) composited Global Ozone Monitoring Experiment (GOME) tropospheric column nitrogen dioxide (TCNO$_2$) under the winter-time NAO finding an increase (decrease) of 3-5 (3-7) $\times$ 10$^{14}$ molecules/cm$^2$ in the NAO- (NAO+) phase over Scandinavia/eastern Europe (UK and France). Thomas and Devasthale (2014), using Atmospheric Infrared Sounder (AIRS) CO at 500 hPa, found that NAO+ and NAO- significantly increased (2.5%) and decreased (4%) CO concentrations over the Nordic countries, respectively. Creilson et al. (2003) investigated the links between the NAO and tropospheric ozone from the Total Ozone Mapping Spectrometer/Solar Backscattered Ultraviolet tropospheric ozone residual (TOR) product (1979-2000). They found the largest correlations between the NAO and TOR in winter/spring, where tropospheric ozone is larger by 3-5 DU (0-2 DU) over central-western Europe (Mediterranean) during the spring-time NAO+ (NAO-). Pausata et al. (2012) found that in December-January-February (DJF) NAO+ significantly increases (90% confidence level) surface ozone by 6-10 ppbv over the UK and northern Europe, while NAO- leads to a decrease of 4-10





ppbv. This is indicative of Atlantic ozone-rich air being transported into Europe under strong westerly flow during NAO+. Under NAO-, the weaker winds are re-oriented over southern continental Europe aiding the accumulation of primary pollutants (e.g. nitric oxide, NO), which acts as a substantial sink of ozone in winter. Bacer et al. (2016) and Christoudias et al. (2012) used modelled artificial CO/aerosol tracers, to find significant negative (positive) correlations between the NAO meteorological fields and composition over Europe (Canada). This highlighted the replacement of European pollution under NAO+ with clean Atlantic air, while pollution accumulated over continental Europe during NAO-.

Here, we build upon previous work by using satellite measurements of tropospheric ozone and upper troposphere - lower stratosphere (UTLS) peroxyacetyl nitrate (PAN) to investigate the impact of the NAO circulation on tropospheric composition. We also analyse satellite measurements of TCNO$_2$ from the Ozone Monitoring Instrument (OMI), which provides higher resolution and more frequent observations over a longer timer period than the GOME record used by Eckhardt et al. (2003). The TOMCAT chemistry transport model (CTM) is utilised to help diagnose these relationships seen in the satellite data in order to understand key processes which govern pollution levels over sources regions and populated areas. Section 2 discusses the observations and model setup, Section 3 describes the links between satellite observed/model composition and the NAO and our discussion/conclusions are presented in Section 4/5.

## 2  Observations and Model

### 2.1  North Atlantic Oscillation Index

Jones et al. (1997) define the North Atlantic Oscillation Index (NAOI) as "the normalised pressure at the southern location (i.e. Gibraltar) minus the normalised pressure at the Icelandic site (i.e. Reykjavik)". The 2006-2015 normalised winter-time (DJF) NAOI time-series, obtained from the Climatic Research Unit (CRU), University of East An-





glia, is plotted in Figure 1a. Here, significant NAO+ and NAO- phases occur when the time-series is greater or less than 1.0 and -1.0 standard deviations (blue dotted lines), respectively. However, the multiple satellite records used in this study to assess the composition-circulation relationships all cover different time periods, so the NAO

5 phases are determined based on their corresponding NAOI time-series. We then compare the satellite composition-circulation relationships with the model relationships for the 2006-2015 period. Satellite data also have periods of missing data, especially in winter, so the season is extended in some cases (e.g. includes November) to build up a sizeable satellite composite for more robust signals.

10 Figures 1b-e show the pressure anomalies, relative to the DJF 2006-2015 average, and winds under both NAO phases at the surface and 10 km derived from European Centre for Medium-Range Weather Forecasts (ECMWF) ERA-Interim data. Under NAO+ (Figure 1b), the Icelandic low pressure system intensifies by 5-10 hPa at the surface. Over the subtropical North Atlantic, surface pressure increases by

15 3-5 hPa, yielding a stronger meridional Atlantic pressure gradient. Therefore, enhanced westerly winds (i.e. storm tracks) peaking over 10 m/s are orientated over north-western Europe. Green polygonned regions show significant differences (99%) between NAO composite pressure and the DJF average using the Wilcoxon Rank Test (WRT, Pirovano et al. (2012)). NAO- (Figure 1c) shows the opposite pattern,

20 with significant positive (negative) pressure anomalies of 5-15 hPa over the Southern Greenland/Iceland (subtropical North Atlantic). The pressure anomaly reversal yields a weaker meridional pressure gradient, slower easterly winds (below 10 m/s) between 50-60°N and southwards shift in the storm tracks. At 10 km, the spatial structure and significance of the pressure anomalies are similar to the surface, but with smaller abso-

25 lute differences (i.e. as pressure decreases with altitude). The more uniform westerly flow (Figure 1d) peaks at approximately 40 m/s. Under NAO- (Figure 1e) the 10 km westerlies are over the sub-tropical North Atlantic with weakened flow (approximately 10 m/s) over the mid-North Atlantic.





## 2.2 Satellite Observations

To investigate the links between tropospheric composition and the NAO, we have used satellite measurements of $TCNO_2$, tropospheric ozone profiles (and sub-columns, 0-6 km) and UTLS PAN. $TCNO_2$ (DOMINO product v2.0; Boersma et al. (2011b)) for 2005 to 2015 comes from OMI, on-board NASA's AURA satellite (2004 - present), with a sun-synchronous overpass of approximately 13.30 local time (LT). OMI is nadir-viewing with a spectral range of 270-500 nm and pixel footprint sizes of 16-23 km and 24-135 km along and across track, respectively (Boersma et al., 2007). A full description of the OMI $NO_2$ retrieval is discussed by Eskes and Boersma (2003). Individual retrievals were screened for poor data quality flags, geometric cloud fraction greater than 0.2 and the OMI row anomalies (Braak, 2010).

Tropospheric ozone measurements used here are from the Tropospheric Emission Spectrometer (TES) on-board the NASA's AURA satellite. TES is an infrared Fourier transform spectrometer that measures thermal emissions over the spectral range of 650 - 2250 cm$^{-1}$ and has a nadir-viewing footprint of 45 km$^2$ (Richards et al., 2013). TES has peak sensitivity to lower tropospheric ozone at approximately 850 hPa (Worden et al., 2013). The TES data has also been screened for poor data quality flags.

The Michelson Interferometer for Passive Atmospheric Sounding (MIPAS) operated onboard ESA's ENVISAT satellite between 2002-2012 and measured many trace gases including PAN in the UTLS. ENVISAT was a sun-synchronous polar-orbiting satellite, which performed 14.4 orbits per day, crossing the equator at about 10:00 am and 10:00 pm LT. MIPAS was a limb-viewing emission spectrometer covering the spectral region between 685 and 2410 cm$^{-1}$ (Fischer et al., 2008), which produced up to 1400 profiles each day. The measurements, in reduced resolution nominal mode, had 27 tangent altitudes per limb scan. The lowermost (uppermost) tangent altitudes ranged approximately from 5 km (70 km) near the poles to 12 km (77 km) at the equator (Wiegele et al., 2012). The data was retrieved from the Karlsruhe Institute for Technology (KIT) for the full satellite mission (full spectral resolution -





V5H, 2002-2004 and reduced spectral resolution - V5R, 2005-2012). The full file type list is 2002: V5H_PAN_20, 2003: V5H_PAN_20, 2004: V5H_PAN_20, 2005: V5R_PAN_220, 2006:V5R_PAN_220, 2007: V5R_PAN_220, 2008: V5R_PAN_220, 2009: V5R_PAN_220, 2010: V5R_PAN_220, 2011: V5R_PAN_220/1 and 2012: V5R_PAN_221.

In terms of satellite errors and uncertainties, the random errors are primarily assessed when compositing different chemical species under the two NAO phases. The systematic errors will cancel considerably when comparing species-NAO composites to their winter-time averages. The averaging of daily data will reduced the random error component by a factor of $1.0/\sqrt{n}$, where $n$ represents the number of days with good quality satellite data. The random errors of the different species under both NAO phases is discussed further in the supplementary material (SM). However, OMI TCNO$_2$, MIPAS PAN 150 hPa and TES lower tropospheric ozone random errors range from approximately 10-40%, peak at 15-20% and are between 10-20% over the North Atlantic and western Europe. Boersma et al. (2004), Glatthor et al. (2007) and Richards et al. (2008) provided detailed discussion on these product uncertainties, respectively.

## 2.3 TOMCAT 3-D Model

TOMCAT is a three-dimensional (3-D) off-line chemistry transport model (CTM; e.g. Chipperfield et al. (1993); Stockwell and Chipperfield (1999); Chipperfield (2006)). ECMWF ERA-Interim meteorological analyses are used to force the model winds, temperature, and humidity (Dee et al., 2011). The standard TOMCAT tropospheric chemistry version uses 82 advected tracers and 229 gas-phase reactions (Emmons et al., 2015), which includes the extended tropospheric chemistry (ExTC) scheme. The VOC degradation chemistry scheme incorporates the oxidation of monoterpenes, C2-C4 alkanes, toluene, ethene, propene, acetone, methanol and acetaldehyde, which was implemented by Monks et al. (2017). The model chemistry scheme includes the Mainz condensed isoprene oxidation mechanism (MIM) (Pöschl et al., 2000). TOMCAT also includes heterogeneous N$_2$O$_5$ hydrolysis using on-line size-resolved aerosol from




the Global Model of Aerosol Processes (GLOMAP) model (Mann et al., 2010). Aerosol types have individual uptake coefficients as parameterized by Evans and Jacob (2005), with the exception of dust which is based on Mogili et al. (2006). Tracer advection by the resolved winds is performed using the scheme of Prather (1986). Subgrid scale transport is performed using the Tiedtke convection scheme (Tiedtke, 1989; Stockwell and Chipperfield, 1999; Feng et al., 2011) and the Holtslag and Boville (1993) parameterization for turbulent mixing in the boundary layer following the method of Wang et al. (1999). Where available, kinetic data are taken from IUPAC (http:www.iupac-kinetic.ch.cam.ac.uk) and for other reactions, we use the Leeds Master Chemical Mechanism (MCM). The model anthropogenic emissions come from the Streets v1.2 inventory, which is a composite of several regional emissions inventories (Emmons et al., 2015). The MACCity inventory (Granier et al., 2011) is used for the natural emissions and biomass burning emissions come from the Global Fire Emissions Database (GFED) v3.1 inventory (Randerson et al., 2013).The model was initialised in December 2005, using a restart (initialisation) file from previous simulations, and run for 2006 to 2015 at the $2.8° \times 2.8°$ spatial resolution (Monks et al., 2017).

## 3 Results

### 3.1 Observations of Tropospheric Composition

#### 3.1.1 Nitrogen Dioxide

OMI TCNO$_2$ was sampled under the winter-time (November - February) NAO+ (Figure 2a) and NAO- (Figure 2b) for 2005-2015. Peak TCNO$_2$ concentrations (over $15 \times 10^{15}$ molecules/cm$^2$) in both phases are over the Po Valley and the Benelux region. Over the UK, source region TCNO$_2$ ranges between 7-13 $\times 10^{15}$ molecules/cm$^2$ and 6-10 $\times 10^{15}$ molecules/cm$^2$ in NAO- and NAO+, respectively. We hypothesise that NAO+ enhanced westerly flow transports NO$_2$ off the UK mainland, as seen by Pope et al. (2014) who




investigated the impacts of cyclonic conditions on UK $TCNO_2$. Figure 2c supports this, highlighting significant negative anomalies of -4 to -2 $\times$ $10^{15}$ molecules/cm$^2$ between the $TCNO_2$ NAO+ composite and 11-year winter-time average. Significant anomalies, shown in the green polygonned regions, are based on the WRT at the 95% confidence level and where composite and winter-time averages $\pm$ their respective random errors (Pope et al., 2015) do not overlap. Systematic errors will cancel when differencing the two $TCNO_2$ composites. NAO- reduces westerly flow across Europe and might be expected to aid $TCNO_2$ accumulation, but there is actually little change in the anomaly field (Figure 2d). Only the Benelux region (3-5 $\times$ $10^{15}$ molecules/cm$^2$) and North Sea (-2.0 to -1.0 $\times$ $10^{15}$ molecules/cm$^2$) show significant anomalies linked to $NO_2$ accumulation and reduced transport off the UK mainland.

### 3.1.2 Peroxyacetyl Nitrate

The MIPAS PAN 200-100 hPa average volume mixing ratio, sampled under NAO+ (DJF) between 2002-2012 (Figure 3a), shows peak (minimum) PAN concentrations of 50-55 (10-20) pptv in the sub-tropical North Atlantic (over Newfoundland and the Canadian Arctic). During NAO- (Figure 3b), PAN concentrations are lower over the sub-tropical Atlantic, but slightly larger over Newfoundland/the Canadian Arctic between 30-45 pptv. PAN concentrations are also larger (approximately 40 pptv) over Iceland/Southern Greenland/Denmark Strait leaving a spatially prominent feature. MIPAS-derived tropopause height (see SM) peaks at 11 km in this region while it is only 9-10 km in the surrounding area (excluding the sub-tropical North Atlantic). There is also an increase in pressure and convergence of winds over this region (Figure 1d) potentially highlighting the impact of NAO- vertical transport of PAN into the UTLS; this is investigated further using TOMCAT (see Section 3.2.2). In Figure 3c under NAO+, peak significant anomalies of 5-15 pptv occur over the sub-tropical Atlantic and Arctic region northwards of 70°N. There are also significant negative anomalies (-5 to -1 pptv) over the Quebec region. Significant anomalies are based on the WRT (95% confidence level) and where the NAO composite and the winter-time averages $\pm$ their uncertainty



ranges do not overlap. Over Iceland/Greenland (sub-tropical North Atlantic and Europe), there are positive (negative) anomalies of 5-15 (-5 to -1) pptv in NAO-.

Figure 4 shows the zonally averaged (90°-20°E) vertical profiles of MIPAS PAN under the two NAO phases. Peak PAN concentrations at 300-250 hPa range between 100 and 130 pptv in both phases, but are larger in NAO- between 30-50°N by 10-20 pptv. However, northwards of 70°N, PAN concentrations between 225 and 100 hPa tend to be larger under NAO+ conditions. Figure 4c shows the NAO+ zonal anomalies relative to the winter-time average (hatched anomalies are insignificant based on the WRT - 95% confidence level). Northwards of 50°N, significant positive anomalies (5-15 pptv) exist between 300-100 hPa. MIPAS derived tropopause height under NAO+ (see SM) is typically higher than in NAO- over the North Atlantic and Europe by 1-2 km. Therefore, the higher tropopause potentially aids vertical transport into the UTLS, which promotes elevated PAN concentrations originating from further down in the troposphere. Southwards of 50°N, positive anomalies occur between 200-100 hPa, while negative anomalies are found between 300-250 hPa. Under NAO- conditions (Figure 4d), there are significant positive anomalies (5-15 pptv) at 300-275 hPa between 60-90°N and they reach up to 125 hPa at 50-70°N coinciding with peak PAN concentrations over Iceland in Figure 3b. Significant negative anomalies (-5 to -1 pptv) exist at 225-50 hPa which coincide with the negative anomalies in Figure 3d over the sub-tropical Atlantic. Between 30-50°N, there is an altitude anomaly dipole reversal with NAO+ showing significant positive (negative) anomalies at 200-100 hPa (300-250 hPa) and NAO- highlighting significant positive (negative) anomalies at 300-275 hPa (225-50 hPa). These patterns are linked to changes to regional circulation patterns under the different NAO phases and will be explored further using TOMCAT in Section 3.2.2.

### 3.1.3 Ozone

Figure 5 shows TES vertical profiles averaged over four regions (Zones 1-4) covering the North Atlantic, between 2005 and 2011, sampled during significant winter-time





(November-February) NAO events. In Zone 1 (UK), TES ozone sampled under NAO+ (red line) is significantly larger (90% - squares and 95% -diamonds) than in NAO- (blue line) by 3-4 ppbv throughout the region of peak sensitivity (900-650 hPa - yellow box). Similar patterns exist in surface ozone measurements from the UK Automated Urban and Rural Network (AURN, DEFRA (2015)). Under NAO+, surface ozone concentrations were significantly higher than in NAO- by 5-10 $\mu$g/m$^3$ across the UK (see SM). The opposite is true for AURN surface NO$_2$ where concentrations across the UK are lower by 5-10 $\mu$g/m$^3$ in NAO+. This supports the hypothesis that NAO+ increases (decreases) ozone concentrations over western Europe (western Atlantic) through enhanced westerly transport of ozone and dispersion of other species involved in its removal (e.g. NO) over Europe. Zone 2 (Newfoundland) has the opposite signal where the NAO- ozone profile is significantly (95%) larger by 2-4 ppbv than the NAO+ profile. Again, this is potentially linked to enhanced westerly ozone transport across the Atlantic towards Europe during NAO+. In Zone 3 (North Atlantic), there are insignificant differences at approximately 900 hPa, however, ozone is significantly greater under NAO- between 875-350 hPa. There are insignificant differences in Zone 4 ozone up to 600 hPa, but NAO+ ozone is significantly larger above this altitude.

## 3.2 Model Composition

TOMCAT has been evaluated in multiple studies (e.g. Monks et al. (2017); Richards et al. (2013)) for NO$_2$, PAN and ozone, which is discussed in detail in the SM. We also have evaluated TOMCAT surface/tropospheric ozone against a range of observations covering western Europe and the North Atlantic. In all cases, TOMCAT has suitable skill to represent these chemical tracers and their responses to the NAO circulation patterns.





### 3.2.1 Nitrogen Dioxide

In NAO+ and NAO- (Figure 6a & b), where TOMCAT has been sampled under the NAO phases in Figure 1a, the model $TCNO_2$ over western Europe ranges between 3 to 9 $\times 10^{15}$ and 6 to over $10 \times 10^{15}$, respectively. Over the UK, NAO+ enhanced westerly flow transports $NO_2$ off the mainland (Figure 6c) with significant negative anomalies of -2.0 to -0.5, relative to the winter-time average. OMI $TCNO_2$ has a similar NAO+ UK signal (2c), but it is less spatially extensive and does not cover as much of continental Europe. In NAO-, OMI (Figure 2d) only shows accumulation of $TCNO_2$ over the Benelux region, while TOMCAT (positive anomalies over $1.5 \times 10^{15}$ molecules/cm$^2$) accumulates $TCNO_2$ over all of continental Europe (Figure 6d).

At the surface, TOMCAT surface $NO_2$ ranges between 0-6 ppbv and 2-8 ppbv in NAO+ and NAO-, respectively. TOMCAT does have a systematic surface $NO_2$ low bias against surface observations (see SM), but this systematic offset is removed when considering anomalies (Figure 8c & d), relative to the winter-time average. TOMCAT surface anomalies typically have similar spatial patterns to the TOMCAT $TCNO_2$, but they are less spatially extensive. Under the NAO+, there are significant negative (positive) anomalies of -0.5 (0.2) ppbv over the UK (North Sea) highlighting the westerly transport of $NO_2$ off the UK mainland. Under NAO-, significant positive anomalies (0.0 to 1.0 ppbv) highlight the accumulation of $NO_2$ from reduced westerly flow across the UK. This is consistent with the AURN results presented in the SM. The model also shows a significant anomaly dipole over Scandinavia which reverses between phases. This, in combination with the reduced spatial impact on surface $NO_2$ compared with the tropospheric pattern, implies that processes above the surface also influence the response of the tropospheric $NO_2$ distribution to the NAO.

Figure 8 shows the TOMCAT $NO_2$ meridional-vertical cross-section at 0°E. Between 35-60°N TOMCAT simulates $NO_2$ concentrations above 1.0 ppbv from 1000 hPa to 900 (800) hPa in NAO+ (NAO-). Negative anomalies (under -0.2 ppbv), relative to the winter-time average, from 1000-900 hPa at 50°N (Figure 8c) show the enhanced





NAO+ westerly flow transporting $NO_2$ throughout the boundary layer away from UK source regions. This $NO_2$ is transported into the North Sea yielding positive biases of 0.1 ppbv northwards of 55°N with vertical ascent into the mid-troposphere (approximately 600-700 hPa). Under NAO- (Figure 8d), there are positive (0.2 pptv) anomalies between 40-55°N as the weakened meridional winds have a southerly flow with ascent (descent) at 65 (40)°N. This highlights reduced $NO_2$ transport from the climatological westerlies aiding accumulation in the lower troposphere (1000-700 hPa). Therefore, processes throughout the lower troposphere over the UK are important in governing the tropospheric column burden during the two NAO phases.

### 3.2.2 Peroxyacetyl Nitrate

At the surface, although PAN has lower concentrations than $NO_2$ in source regions, it has a longer lifetime resulting in more significant responses to the seasonal average under the different NAO phases. Under NAO+ (Figure 9a), TOMCAT surface PAN peaks between 200-220 pptv over the Western Atlantic. Over Europe, PAN ranges between 150-170 pptv as, like $NO_2$, enhanced westerly flow transports PAN away from western European source regions replacing it with cleaner sub-tropical North Atlantic air (100-150 pptv). Through reduced transport, NAO- conditions aid pollutant accumulation over continental Europe with PAN concentrations of 190 to over 300 pptv. The NAO+ TOMCAT PAN anomalies (Figure 9c), relative to the winter-time average, highlight reduced concentrations of -50 to -20 pptv over continental Europe, while in the western North Atlantic there are no significant anomalies. This infers similar transport processes to the winter-time average resulting in minimal PAN changes, yet NAO- (Figure 9d) weakens/reverses the winds yielding significant negative anomalies of -20 to -10 pptv. Therefore, westerly flow, similar under NAO+ and average winter-time conditions, aids the long-range transport of PAN from North America. As NAO- interrupts this transport pathway, there is a significant decrease in background PAN.

We now investigate whether TOMCAT reproduces the MIPAS UTLS PAN patterns under the NAO phases, despite the slightly different time periods. Previous studies



(e.g. Emmons et al. (2015); Pope et al. (2016)) have shown that TOMCAT PAN compares reasonably well with aircraft observations, but there is a systematic difference between TOMCAT and KIT MIPAS PAN (see SM). Therefore, we primarily focus on the anomalies, relative to the winter-time average, under the NAO phases as this systemic difference is removed. TOMCAT PAN 200-100 hPa average peaks (over 50 pptv), in NAO+, over the western subtropical North Atlantic (Figure 10a). The south-westerly flow (approximately 30 m/s) at this altitude transports PAN across the Atlantic reaching 35 pptv over Iberia. However, at approximately 0°E, a southerly shift in the winds over the Mediterranean leads to lower continental Europe PAN concentrations (20-30 pptv). Northwards of 70°N, the flow (20-30 m/s) accumulates PAN over the Arctic region (20-24 pptv). Similar spatial patterns are seen in MIPAS with peak PAN concentrations over the westerly sub-tropical Atlantic, minimum PAN over Canada/Hudson Bay and elevated PAN in the Arctic region. However, MIPAS PAN absolute concentrations are systematically larger than TOMCAT (see SM). Vertical transport will also have an important impact as the higher MIPAS-derived tropopause height in NAO+ (see SM) allows for the uplift of PAN into this altitude range. In NAO-, peak PAN (over 40 ppbv) occurs in the sub-tropical Atlantic where the winds are predominately zonal (westerly) yielding lower PAN concentrations (15-25 pptv) over the mid-North Atlantic. Continental Europe PAN concentrations have decreased (10-20 pptv), as north-westerly flow is transporting cleaner Arctic air masses into the region. PAN accumulation over Iceland and Southern Greenland (25 ppbv) correlates with the large UTLS pressure increase shown in Figure 1d. Figure 10d highlights the significant enhancement of PAN over Iceland/Southern Greenland with positive anomalies, relative to the winter-time average, of 5-10 pptv. Again, the MIPAS-derived tropopause height in NAO- peaks in this region (approximately 11 km - see SM) suggesting sufficiently strong vertical transport of tropospheric air masses. In NAO-, as the strong westerly flow in NAO+ (Figure 10b) has shifted equator-wards, there are significant negative anomalies under -15 pptv across the North Atlantic, which match the MIPAS equivalent in Figure 3d. There are some similarities between the TOMCAT (Figure 10c) and MIPAS (Figure 3c) NAO+ PAN





anomalies with increased PAN in the eastern Arctic. However, TOMCAT simulates insignificant positive anomalies (0-2 pptv) over the North Atlantic between 35-45°N while MIPAS has significant positive anomalies (10 ppbv). TOMCAT (MIPAS) also simulates (observes) significant negative anomalies over the UK/western North Atlantic (northern Canada), which are not seen (simulated) by MIPAS (TOMCAT).

Figure 11 shows the zonal average (90°W-20°E) meridional-vertical TOMCAT PAN distribution under both NAO phases. Between 1000 and 600 hPa, PAN concentrations are above 200 pptv apart from in the region 30-40°N. From 400-200 hPa, there is a sharp PAN decrease to less than 30 pptv. The vertical PAN profile between 30-40°N differs from other latitude bands with PAN peaking at 150-200 pptv from 1000-700 hPa and then 60-100 pptv up to 200 hPa. Between 200-100 hPa, PAN concentrations (30-60 pptv) are larger than other latitude bands linked to the higher tropopause (also observed by MIPAS - see SM). The significant decreases in surface PAN over Europe from NAO+ enhanced westerly flow (Figure 9c) occur throughout the troposphere with negative zonal anomalies of -10 to -3 pptv (Figure 11c). Above the tropopause (dashed green line), strong vertical (w winds are scaled by $10^4$ for clarity) - meridional transport accumulates PAN (positive anomalies of 3-6 pptv) in Arctic UTLS. Under NAO- (Figure 11d), there are positive (5-10 pptv) and negative (-10 to -5 pptv) anomalies between 30-50 and 60-90°N throughout the troposphere. The surface patterns (Figure 9d), where reduced transport aids PAN accumulation over Europe, appear to account for this zonal tropospheric pattern. Between 30-50°N, there is limited meridional flow aiding PAN accumulation over Europe in NAO-. The vertical flow contributes to positive anomalies (3-5 pptv) propagating into the UTLS consistent with PAN accumulation shown in Figure 10d between 50-70°N.

### 3.2.3 Ozone

TOMCAT surface ozone under NAO+ (Figure 12a) peaks at approximately 28-30 ppbv over sub-tropical and western North Atlantic co-located with the enhanced westerlies. Over continental Europe, ozone concentrations are significantly larger (1-2 ppbv - Fig-





ure 12c) than the winter-time average ranging between 13-22 ppbv. This matches a similar pattern in the observations; AURN surface ozone was significantly higher over the UK under NAO+ than NAO- (SM) and TES lower tropospheric ozone (Zone 1, Figure 5) was larger under NAO+. In Zone 2, TES lower tropospheric ozone was higher under NAO-, which correlates with the surface TOMCAT pattern. Pausata et al. (2012) found similar patterns with significant positive (negative) correlations over Europe (western North Atlantic) between surface ozone and the NAOI in DJF. Under NAO- conditions, TOMCAT ozone concentrations are consistent across the North Atlantic (28-30 ppbv) as the weakened/reversed westerlies limit the transport of ozone-enriched Atlantic into Europe yielding lower concentrations of 13-16 ppbv. Over the western North Atlantic (Europe), ozone concentrations (Figure 12d) have increased (decreased) with significant positive (negative) anomalies of 2-3 ppbv (-3 to -1 pptv). Again, Pausata et al. (2012) presented similar results which also match TES and AURN ozone observations (see SM). TOMCAT tropospheric column ozone (not shown here) also showed similar anomalies.

At the surface, the PAN and $NO_2$ spatial anomalies are anti-correlated with ozone, so UTLS ozone (Figure 13) was investigated to see if this relationship was consistent at higher altitudes. TOMCAT 200-100 hPa average ozone, sampled under NAO+, ranges from 800 to 1000 ppbv northwards of 60°N but decreases towards the subtropical North Atlantic with minimum concentrations of 150-200 ppbv. A similar pattern occurs under NAO- except for the ozone reduced air mass (500-700 pptv) stretching from approximately 45-65°N along 15-45°W. Higher ozone concentrations (800-1000 ppbv) also propagates further south in NAO- on either side of the Atlantic surrounding the reduced ozone limb. The UTLS ozone anomalies (Figure 13c & d) are also anti-correlated with the PAN. Whereas PAN has positive (mainly non-significant) anomalies across the Atlantic basin in NAO+, there are significant negative ozone anomalies (-100 to -30 ppbv). Over the UK/eastern North Atlantic, TOMCAT simulates significant negative PAN anomalies, while the ozone anomalies are positive (non-significant). This anti-correlation is more prominent under NAO-, where significant TOMCAT ozone


anomalies (50-200 ppbv) exist over the mid-North Atlantic and Europe, while significantly negative for PAN. As shown in Figures 10 and 11, tropospheric positive PAN anomalies propagate into the UTLS over Iceland/Southern Greenland, but the ozone anomalies are significantly negative (-150 to -50 ppbv). Potential reasons for the PAN-ozone anti-correlation include the air mass origin or the PAN ($NO_x$)-ozone chemistry. The thermal decomposition of PAN forms the peroxyacetyl radical and $NO_2$, which is an UTLS ozone sink (i.e. conversion of $NO_2$ to NO and then reaction with ozone) while a tropospheric ozone source in the presence of volatile organic compounds (Richards et al., 2013). However, lower UTLS temperatures (i.e. around 250 K) yield a PAN lifetime of several months (Singh, 1987) and a less likely factor in the PAN/$NO_x$-ozone anomaly anti-correlations. Cohen et al. (1994) show that UTLS ozone-$HO_x$ chemistry is a more significant sink pathway for ozone, however, there is no clear correlation between the NAO $HO_2$ and ozone anomalies. Methane, a good air mass tracer due to its approximate 9-year lifetime (e.g. McNorton et al. (2016)) and anthropogenic source, was sampled under the NAO phases (not shown) and highlighted similar anomaly patterns to PAN, again anti-correlated with ozone. Therefore, PAN and methane (ozone) act as signatures for the transport of polluted (clean) air masses in the troposphere for the different NAO phases.

Figure 14 shows the TOMCAT ozone cross-section at 0°E, similar to $NO_2$ in Figure 8. Under both NAO phases, lower-tropospheric (UTLS - above 300 hPa) ozone ranges between 25-35 (above 100) ppbv. Meridionally, there is a decreasing poleward lower tropospheric ozone gradient, while in the UTLS peak (minimum) concentrations are at the pole (30°N). The anomalies, as discussed above, are anti-correlated with $NO_2$. In NAO+, there are small positive (negative) anomalies over the UK (North Sea) consistent with ozone-enriched air transported into the UK from the North Atlantic and ozone loss downwind due to source region $NO_x$ transport, which propagate up to approximately 600 hPa. The positive anomalies in the UTLS at 60°N are consistent with ozone accumulation seen in Figure 13. Under NAO- conditions, the negative anomalies (approximately -3 ppbv) between 45-65°N and 1000-700 hPa are linked to UK $NO_x$



accumulation (Figure 7d). Atmospheric downwelling leads to UTLS ozone (positive anomalies over 3 ppbv) propagation into the mid-troposphere at 40-50°N. At high latitudes, small positive anomalies throughout the troposphere, which are anti-correlated with PAN, show clean air transport around the UTLS Icelandic-Southern Greenland anticyclone in which PAN accumulates.

A second TOMCAT ozone cross-section, at 56.25°W (Figure 15) has similar absolute ozone concentrations to the 0°E cross-sections, but the anomalies highlight important differences. In NAO+, both cross-sections have similar lower tropospheric ozone anomalies except at 50°N (Figures 14c and 15c) with positive anomalies (approximately 1 ppbv) over the UK region and near 0 ppbv over the western North Atlantic. Under NAO- conditions, there are positive anomalies (Figure 15d, 1-3 ppbv) between 1000-600 hPa and 45-70°N, but the eastern cross-section (Figure 14d) highlights negative anomalies in this region (-3 to -1 pptv). While there is downwelling of stratospheric ozone in the eastern cross-section into the upper troposphere during NAO-, the western cross-section has upwelling of ozone reduced air into the UTLS with negative anomalies of less than -5 ppbv. Overall, in the lower troposphere, the TOMCAT cross-section anomalies support the signals in the TES data. Over the UK (Zone 1, Figure 5, and eastern cross-section, Figure 14), lower tropospheric ozone is larger (lower) than the winter-time average under NAO+ (NAO-), while the opposite occurs in western North Atlantic (Zone 2 and western cross-section, Figure 15).

## 4 Discussion

Analysis of satellite-observed and model-simulated atmospheric composition sampled under the different winter-time NAO phases clearly highlights the importance of transport, both horizontal and vertical, for variability in concentrations of the air pollutant species investigated. At the surface and in the lower troposphere, enhanced westerly flow in NAO+ influences primary pollutant concentrations (e.g. $NO_2$) over Europe as they are transported away from source regions and replaced by clear Atlantic air





masses. As $NO_2$ has a short lifetime of several hours, there is little impact of the NAO circulation in the upper troposphere where $NO_2$ concentrations are much lower. Under NAO-, the reduced westerly flow significantly aids the accumulation of $NO_2$ at levels between the surface and approximately 600 hPa. This is important for air pollution levels over source regions which are predominately highly polluted. Ozone has the opposite signal to $NO_2$ in the lower troposphere where NAO+ replaces primary polluted air (e.g. high $NO_x$ content) over Europe with ozone-enriched Atlantic air masses. The high ozone content of these air masses is linked to ozone formed downwind of primary pollution from North America and decreased levels of ozone depleting gases (e.g. NO, when photochemical and OH activity are slower) over Europe. Over North America and the western North Atlantic, NAO+ and NAO- show significant decreases and increases in tropospheric ozone, respectively, as the NAO+ enhanced westerlies transport ozone-enriched air masses towards Europe, while NAO- weakens this transport pathway resulting in elevated ozone concentration in the region from North American pollution outflow.

In the UTLS, the spatial distribution of PAN is heavily influenced by both horizontal and vertical transport. In NAO+, as shown by MIPAS (see SM) and Figure 11, the tropopause height is elevated enhancing the vertical transport of PAN into the UTLS over the Arctic and sub-tropical North Atlantic. UTLS horizontal winds also contribute to these elevated PAN concentrations as strong winds (e.g. 30 m/s) help accumulate PAN in the Arctic. In NAO-, poleward flow from the sub-tropical North Atlantic has weakened leading to a decrease in PAN over the North Atlantic and their is no longer the accumulation of Arctic PAN. However, the UTLS anticyclone (as seen in Figure 1e) shows a clear accumulation of PAN in the UTLS over southern Greenland/Iceland linked to vertical transport of PAN from the pollutant lower troposphere over Europe (reduced westerlies in NAO- allow the accumulation of PAN and $NO_x$). Ozone in the troposphere and UTLS is anti-correlated with PAN, which we show to be transport dominated, highlighting regions of air mass intrusions from the troposphere into the stratosphere and vice versa. For instance, in the UTLS Icelandic/Southern Green-





land anticyclone ozone is significantly reduced while PAN is enhanced. In winter, as photochemical activity and reaction with OH is reduced, polluted air masses with high NO$_x$ content will yield low ozone and high PAN concentrations, respectively. However, stratospheric intrusions into the upper troposphere (e.g. Figure 14d between 40-60°N) have a high ozone content but low PAN concentrations as their is limited production of PAN in this part of the atmosphere.

## 5  Conclusions

Overall, this study has successfully shown that satellite data, despite the large uncertainty in individual retrievals, can be an important observational data source to investigate the response of atmospheric composition, primarily in the troposphere, to changes in atmospheric circulation such as the North Atlantic Oscillation (NAO). We have built upon several previous studies (e.g. Eckhardt et al. (2003); Thomas and Devasthale (2014)) by using observations of tropospheric ozone and upper troposphere - lower stratosphere (UTLS) peroxyacetyl nitrate (PAN). We also extend the analysis of Eckhardt et al. (2003) by using tropospheric column nitrogen dioxide (TCNO$_2$) from the Ozone Monitoring Instrument (OMI), which has higher resolution and sampling and longer record than the Global Ozone Monitoring Experiment (GOME).

Our results, supported by simulations from the chemistry transport model (CTM) TOMCAT, show that the different circulation patterns between NAO phases have a significant impact primarily on European tropospheric composition. Under the positive phase of the NAO (NAO+), enhanced westerly flow in the lower troposphere replaces polluted European air masses (e.g. high content of nitrogen oxides - NO$_x$) with cleaner Atlantic air masses. However, tropospheric ozone concentrations are significantly enhanced in this phase as ozone formed downwind of North American pollution outflow are transported into Europe, as well as the dispersion of ozone sink gases (e.g. nitric oxide, NO, when ozone formation is slower from decreased photochemical and OH activity). In the NAO negative phase, the opposite occurs with lower ozone con-





centrations and accumulation of primary pollutants from reduced westerly flow, which are important for air quality levels. We also show that NAO circulation is important for UTLS composition as polluted air masses (i.e. high PAN content) can propagate to this altitude. For instance, the vertical transport and accumulation of PAN in the NAO-anticyclone over Iceland/Southern Greenland. Therefore, the NAO, depending on its phase strength, can be an important driver of winter-time atmospheric composition and air quality across Europe.

*Acknowledgements.* This work was supported by the NERC National Centre for Earth Observation (NCEO). TOMCAT modelling development was supported by the National Centre for Atmospheric Science (NCAS). Simulations were performed on the national Archer and Leeds ARC HPC systems. We acknowledge the use of the Tropospheric Emission Monitoring Internet Service (TEMIS) OMI tropospheric column $NO_2$ data (DOMINO product v2.0), which is available at www.temis.nl/airpollution/no2. We also acknowledge the use of data from AURN supported by the Department of Environment, Food and Rural Affairs (DEFRA), which is available at www.uk-air.defra.gov.uk/networks/network-info?view=aurn, and Lerwick ozonesounde data provide by the World Ozone and Ultraviolet Radiation Data Centre (http://woudc.org/). We also thank the Climatic Research Unit, University of East Anglia for the use of their North Atlantic Oscillation Index data, which can be found at www.crudata.uea.ac.uk/cru/data/nao. Finally, we acknowledge the use of the TES ozone data provided by NASA's JPL (www.reverb.echo.nasa.gov/reverb) and MIPAS data from KIT (https://www.imk-asf.kit.edu/english/308.php).

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



**Fig. 1.** a) The Climate Research Unit (CRU), University of East Anglia, winter-time (December-January-February, DJF) normalised North Atlantic Oscillation (NAO) index for 2006 to 2015. Red and blue lines show the zero and one standard deviation thresholds, where NAO index values outside this range are classed as significant phases. Panels b) and c) show the NAO positive and negative phase surface pressure anomalies (ERA-Interim data) relative to the winter-time average. Panels d) and e) show the same as b) and c), but for 10 km altitude. Wind vectors are over-plotted for the respective NAO phases and altitudes. Green polygonned regions highlight significant differences at the 99% confidence level based on the Wilcoxon Rank Test (WRT).





**Fig. 2.** Mean Ozone Monitoring Instrument (OMI) tropospheric column $NO_2$ ($10^{15}$ molecules/m$^2$) averaged between 2005 and 2015 under the CRU winter-time (November-December-January-February, NDJF) NAO index. Panel a) is column $NO_2$ sampled under the positive NAO phase, b) is column $NO_2$ sampled under the negative NAO phase, c) shows the column $NO_2$ positive NAO phase anomaly relative to the winter-time average and d) is the column $NO_2$ negative NAO phase anomaly relative to the winter-time average. Green polygonned regions highlight significant differences at the 95% confidence level based on the WRT.





**Fig. 3.** Michelson Interferometer for Passive Atmospheric Sounding (MIPAS) peroxyacetyl nitrate (PAN; pptv) averaged between 200 to 100 hPa for 2002-2012. Panel a) shows PAN sampled under the winter-time (DJF) NAO positive phase, b) shows PAN sampled under the winter-time NAO negative phase, c) shows the positive NAO phase anomaly relative to the winter-time average and d) shows the negative NAO phase anomaly relative to the winter-time average. Green polygonned regions highlight significant differences at the 95% confidence level based on the WRT.




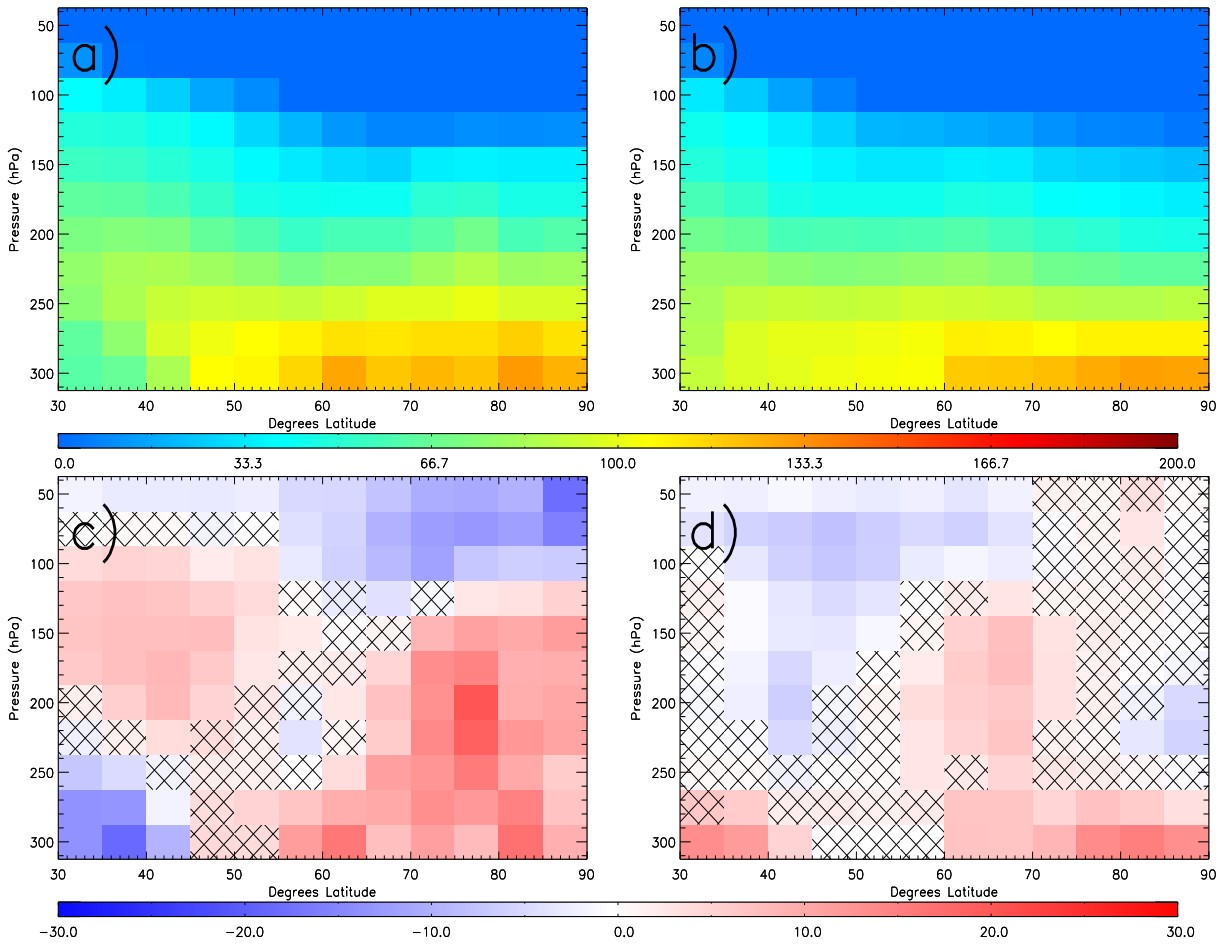

**Fig. 4.** MIPAS PAN (pptv), 2002-2012, zonally averaged (90°W to 20°E) for a) the winter-time (DJF) NAO positive phase, b) the winter-time NAO negative phase, c) the positive NAO phase anomaly relative to the winter-time average and d) the negative NAO phase anomaly relative to the winter-time average. Black hatched regions highlight insignificant differences at the 95% confidence level based on the WRT.





**Fig. 5.** Tropospheric Emission Spectrometer (TES) ozone profiles, averaged over four regions (Zone 1: 12°W-6°E, 48-62°N; Zone 2: 45-65°W, 45-65°N; Zone 3: 10-40°W, 40-50°N; Zone 4: 10-40°W, 55-65°N), between 2005 and 2011, sampled under the winter-time (NDJF) positive (red line) and negative (blue line) phases. Horizontal lines show the satellite uncertainty range, the yellow box highlights the region of peak TES sensitivity to lower tropospheric ozone and the dotted lines show the profile averages +/- their respective standard deviations. Squares and diamonds show where the ozone profiles sampled under each NAO phase are significantly different from each other at the 90% and 95% confidence levels based on the WRT.



**Fig. 6.** TOMCAT troposphere column $NO_2$ ($10^{15}$ molecules/cm$^2$) averaged between 2006 and 2015 sampled under the winter-time (DJF) NAO. Panel a) NAO positive phase, b) NAO negative phase, c) shows the NAO positive phase anomaly relative to the winter-time average and d) shows the negative NAO phase anomaly relative to the winter-time average. Wind vectors show the horizontal 10m winds and the red, green and blue contours represent 980, 1000 and 1020 hPa surface pressure. Green polygonned regions in c) and d) highlight significant differences at the 95% confidence level based on the WRT.



**Fig. 7.** TOMCAT surface NO$_2$ (ppbv) averaged between 2006 and 2015 sampled under the winter-time (DJF) NAO. Panel a) NAO positive phase, b) NAO negative phase, c) shows the NAO positive phase anomaly relative to the winter-time average and d) shows the negative NAO phase anomaly relative to the winter-time average. Wind vectors show the horizontal 10m winds and the red, green and blue contours represent 980, 1000 and 1020 hPa. Green polygonned regions in c) and d) highlight significant differences at the 95% confidence level based on the WRT.







**Fig. 8.** TOMCAT $NO_2$ (ppbv) cross section at $0°E$ averaged between 2006 and 2015 sampled under the winter-time (DJF) NAO. a) NAO positive phase, b) NAO negative phase, c) shows the NAO positive phase anomaly relative to the winter-time average and d) shows the negative NAO phase anomaly relative to the winter-time average. Green dashed lines represents the dynamical tropopause. Wind vectors represent the cross section ($0°E$) meridional and vertical (scaled by $10^4$) wind vectors.





**Fig. 9.** TOMCAT surface PAN (pptv) averaged between 2006 and 2015 sampled under the winter-time (DJF) NAO. Panel a) NAO positive phase, b) NAO negative phase, c) shows the NAO positive phase anomaly relative to the winter-time average and d) shows the negative NAO phase anomaly relative to the winter-time average. Wind vectors show the horizontal 10m winds and the red, green and blue contours represent 980, 1000 and 1020 hPa surface pressure. Green polygonned regions in c) and d) highlight significant differences at the 95% confidence level based on the WRT.







**Fig. 10.** TOMCAT PAN (pptv) averaged between 200 to 100 hPa for 2006-2015 sampled under the winter-time (DJF) NAO. Panel a) NAO positive phase, b) NAO negative phase, c) shows the NAO positive phase anomaly relative to the winter-time average and d) shows the negative NAO phase anomaly relative to the winter-time average. Wind vectors show the horizontal 200-100 hPa winds. Green polygonned regions in c) and d) highlight significant differences at the 95% confidence level based on the WRT.





**Fig. 11.** TOMCAT zonally averaged (90°W to 20°E) PAN (pptv) between 2006 and 2015 sampled under the winter-time (DJF) NAO. Panel a) NAO positive phase, b) NAO negative phase, c) shows the NAO positive phase anomaly relative to the winter-time average and d) shows the negative NAO phase anomaly relative to the winter-time average. Green dashed lines represents the dynamical tropopause. Wind vectors represent the zonally averaged (90°W to 20°E) meridional and vertical (scaled by $10^4$) wind vectors.





**Fig. 12.** TOMCAT surface ozone (ppbv) averaged between 2006 and 2015 sampled under the winter-time (DJF) NAO. Panel a) NAO positive phase, b) NAO negative phase, c) shows the NAO positive phase anomaly relative to the winter-time average and d) shows the negative NAO phase anomaly relative to the winter-time average. Wind vectors show the horizontal 10m winds and the red, green and blue contours represent 980, 1000 and 1020 hPa surface pressure. Green polygonned regions in c) and d) highlight significant differences at the 95% confidence level based on the WRT.



**Fig. 13.** TOMCAT ozone (ppbv) averaged between 200 to 100 hPa for 2006-2015 sampled under the winter-time (DJF) NAO. Panel a) NAO positive phase, b) NAO negative phase, c) represents the NAO positive phase anomaly relative to the winter-time average and d) represents the negative NAO phase anomaly relative to the winter-time average. Wind vectors show the horizontal 200-100 hPa winds. Green polygonned regions in c) and d) highlight significant differences at the 95% confidence level based on the WRT.





**Fig. 14.** TOMCAT ozone (ppbv) cross section at 0°E/W averaged between 2006 and 2015 sampled under the winter-time (DJF) NAO. Panel a) NAO positive phase, b) NAO negative phase, c) shows the NAO positive phase anomaly relative to the winter-time average and d) shows the negative NAO phase anomaly relative to the winter-time average. Green dashed lines represents the dynamical tropopause. Wind vectors represent the cross section (0°E) meridional and vertical (scaled by $10^4$) wind vectors.



**Fig. 15.** TOMCAT ozone (ppbv) cross section at 56.25°W averaged between 2006 and 2015 sampled under the winter-time (DJF) NAO. Panel a) NAO positive phase, b) NAO negative phase, c) shows the NAO positive phase anomaly relative to the winter-time average and d) shows the negative NAO phase anomaly relative to the winter-time average. Green dashed lines represents the dynamical tropopause. Wind vectors represent the cross section (56.25°W) meridional and vertical (scaled by $10^4$) wind vectors.