# Peer review of "Influence of the North Atlantic Oscillation on European tropospheric composition: an observational and modelling study"

_Atmospheric Chemistry and Physics, 2017_

## Referee Comment (RC1) · Anonymous Referee #1 · 16 Jan 2018

acp-2017-979
Influence of the North Atlantic Oscillation on European tropospheric composition: an observational and modelling study
Richard J. Pope et al.

In this paper, the authors combine multiple satellite observations with CTM simulations to characterise the overall distribution of 3 chemical species, $NO_2$, PAN and $O_3$. Whilst the presentation of satellite observations and CTM simulations is clear, the overall aim of the study is too generic – "to investigate the impact of the NAO circulation on tropospheric composition". As a result, the main conclusion of the paper is also very general and does not, in its current form, constitute a new finding. I recommend that this paper undergoes significant major revisions.

General comments
1. Abstract and introduction: The structure of the introduction does not move on from what is known already, towards the issue that remains unknown and will be studied. As a result, it is not possible to determine the innovative aspect of this research. The authors need to state clearly which gap in the scientific knowledge this paper aims to fill and set out some specific research questions or hypothesis.
2. Results: The authors have chosen to present the results in 2 sections. First the observations of tropospheric composition (section 3.1) and second the model simulations of tropospheric composition (section 3.2). Currently, use of the CTM simulations to 'diagnose the relationships seen in the satellite data' relies on the CTM composite wind fields. A similar result could be achieved using re-analysis wind fields. What is the additional benefit of using a complex CTM?
3. The main conclusion is that the NAO is an 'important driver of winter-time atmospheric composition across Europe'. As the authors themselves state in the introduction, several previous studies have reached the same conclusion and therefore this is not a novel result. Please can the authors focus their research question and use the high quality datasets available to them to answer it.

Specific comments
1. Page 2 line 16: 'altitude' is usually used as a vertical distance measurement but here it could be interpreted as a latitudinal distance. Can this be re-worded to make its use clearer?
2. Page 2 line 18: The authors refer to transport from the troposphere into the UTLS. As part of the troposphere is in the UTLS by definition, should troposphere be mid-troposphere here?
3. Page 3 line 10: Here and elsewhere the authors refer to enhanced westerly circulation as 'storm tracks'. Of course, the low-level eddy driven westerlies can be used as a proxy for the storm track but it should be stated that this is the assumption here.
4. Page 3 line 12: 're-orientation' should be reorientation I think.
5. Page 4 line 10: The authors here claim that they are building on previous work but it is not clear from the introduction specifically what research question(s) they are addressing (see also general comment 1).
6. Page 4 line 14: What is the advantage of higher resolution and more frequent observations? How does this allow the authors to discover that was not known before?
7. Page 4 line 24: What time period is used to normalise the pressure data? Also, why does the study focus on wintertime only? In some situations November data is included, but not in all, why isn't the same period used for all of the data?

8. Page 5 line 28: The 10km wind composites during different phases of the NAO are included in figure 1 but only briefly referred to in the text. Why were these figures included and how do they relate to the subsequent analysis?

9. Page 6 section 2.2: This section details the satellite observations used in the study. However, as the authors have not motivated their decision to focus on PAN, $NO_2$ and $O_3$ in the introduction, it reads like a list of available data sources, rather than the necessary data needed to answer the research question posed. Why are these observations rather than others used in the paper?

10. Page 7 section 2.3: This section appears to contain a lot of information that is not necessary to interpret the results or to reproduce the study. For example, the representation of many chemical species not used in this study are described. Please can this be re-written to focus on the information needed to support the results and conclusion of this specific study.

11. Page 8 line 20: How many satellite overpasses are used to create figure 2. Is the noisy data a result of the sampling frequency? If the satellite data were averaged over larger spatial areas would this help with the signal to noise problem?

12. Page 9 line 11: Do the authors have a physical mechanism to explain the lack of significant anomalies in fig 2d? The equivalent CTM simulation (fig 7d) shows significant positive anomalies across western Europe. What is the reason for this difference? Can the CTM be used to understand better the lack of relationship seen in the observations?

13. Page 10 line 13 and page 14 line 16: How does the higher tropopause aid vertical transport of PAN into the UTLS? Is the UTLS a region that moves with the tropopause or is it defined to be a fixed altitude region?

14. Page 14 line 29: Here the authors claim that the CTM PAN and satellite observed PAN have 'some similarities'. Comparison of figures 3c and 10c, and 4c and 11c, show very different spatial patterns. What is the reason for this? Does this mean that the CTM cannot be used to diagnose the relationships seen in the observations as they cannot reproduce the broad features observed during different phases of the NAO?

15. Page 20 line 8: Please remove 'successfully' from this sentence. It is up to the reader, not the authors to judge the success of the paper.

---

## Referee Comment (RC2) · Anonymous Referee #2 · 23 Jan 2018

General comments:
* * *
The authors used satellite observations and the TOMCAT chemistry transport model (CTM) to investigate the influence of the winter-time North Atlantic Oscillation (NAO) phases on the tropospheric concentrations of NO2, PAN, Ozone over the North Atlantic and western Europe.

Though the methods are scientifically sound, the authors fail to properly motivate the study and it is not clear what is the underlying purpose other than to assertain what was presented in numerous other studies that the manuscript cites. The fidnings and

conclusions of the manuscript in my view do not offer any new general implications for atmospheric science. The manuscript fails to expand beyond what is readily established in the literature or introduce sufficiently novel methods or techniques, other than perhaps some some incremental improvements.

My recommendation is that this paper undergoes major revisions before publication in ACP to: properly motivate the choice of remote sensing products, explain the need and purpose of supplementing the analysis of observations with model simulations, and to include more detailed discussion of the scientific implications of the outcomes.

Specific comments:

——————————

p.4 l.24: It's understood that November is sometimes included in the seasonal data to increase low statistics. However, to aid intercomparison wouldn't it be better to be consistent in all cases?

p.5 l.1: "significant" NAO+ and NAO- phases may be misconstrued by the reader. Propose to change to "high and low" as is usually found in the literature.

p.5. l.28: What is the point of including wind vectors in the Figs? They are not referred to in the discussion/analysis.

p.7.l.7: Please clarify what is meant by "The systematic errors will cancel considerably when comparing species-NAO composites to their winter-time averages". Why is that so?

p7.l.14: "range from approximately 10-40%, peak at 15-20% and are between 10-20%" meaning needs to be made clearer. Is that repsectively for OMI/MIPAS/TES?

Sec. 2.3: The section on the TOMCAT model has to be reworked to limit to information that is relevant to the present study.

p.10 l.12: Any reference as to why the higher tropopause potentially aids vertical transport into the UTLS?

p.11 l.21: "We also have evaluated TOMCAT surface/tropospheric ozone against a range of observations. In all cases, TOMCAT has suitable skill to represent these chemical tracers and their responses to the NAO circulation patterns.". Is there any appropriate reference? OR Perhaps include the comparison in the SM?

The NO2 has a much short lifetime in the atmosphere than the timespan considered here and is heavily dependent on emissions. It is not convincing that the randomly scattered green regions of significance in Fig. 2 allow or support any generalised conclusions for NO2 concentrations to be influenced by a seasonal teleconnection pattern.

Technical corrections:

—————————————

p.6 l.3: change to "measurements of total column NO2 (TCNO2)"

p.6 l.4: UTLS PAN -> "UTLS Peroxyacetyl Nitrate (PAN)"

p.7 l.1 Printing the full file type list in the manuscript text is unecessary and extraneous. Either refer to it (or move details in a supplement).

p.7 l.6 and elsewhere "random error" -> "statistical uncertainty"

p.8 l.8: Cite IUPAC website as refence

Fig.3: It's very hard for the reader to discern the green lines and areas of significance on top of dashed lines in panels c) d). Perhaps make bolder?

Sec. 3.2 Title "Model Composition" -> "Model Results"

p.12 l.4,6: Missing TCNO2 units

p.14 l.14: larger -> higher

Fig.5 Squares and diamonds are cluttered over the errors. Perhaps a different representation using horizontal bands and removal of the yellow sensitivity band may be clearer.

[Figure]

---

## Referee Comment (RC3) · Anonymous Referee #3 · 8 Feb 2018

I was going to write a detailed review, but, I noticed that the other two reviewers have already broadly pointed out the concerns I had when I was reading the manuscript. In my opinion, the focus of the study is misplaced and the message is lost in trying to demonstrate/show different things instead of focussing on one particular topic. For example, it is known since few decades that NAO has a strong influence on northern European pollution variability. However, that does not rule out that one should not study this any more as our understanding of the processes and the tools constantly improves. So, I will not criticize the main motivation behind this study, however, I do feel that the authors should have focussed on either revealing/discussing a new mechansim or complimenting the existing ones. The way the study has concluded is too vague

to delineate precisely what new knowledge has been gained. May be, the authors should take a step back and re-think what their results really convey. I also, have a few other points that the authors might consider improving. 1. I do not understand why the authors chose these three species, TCNO2, O3 and PAN and what is their interdependency and why should we study their covariability. 2. The selection of the regions is not properly motivated (Fig.5) 3. It is not clear what have we actually learned from using the model in addition to observations. What is the exact process that was revealed by modeling that was not known before.

---

## Author Comment (AC1) · 5 May 2018

**Response to Reviewer Comments on Manuscript "Influence of the North Atlantic Oscillation on European tropospheric composition: an observational and modelling study"**

We thank the reviewers for their comments and suggestions for the manuscript. The reviewer comments are given below in black text, followed by our responses in red text and any additions to the manuscript in blue text. References to page and line numbers are based on the ACPD version of the manuscript.

**Firstly, as all three reviewers had similar "general comments", we have prepared responses to these points applicable to all the reviewers. These are:**

1.1. Why is the work novel?

Several studies have indeed looked at the interaction of the NAO and composition using datasets (e.g. satellite or surface) and models. However, many of these studies were performed 10-15 years ago and new satellite missions (see point 1.2) allow us to build upon previous work (e.g. Eckhardt et al., 2003; Creilson et al., 2003) to see if these relationships seen in the past are still valid and if newer more sophisticated instruments see similar or different patterns. Secondly, some studies (e.g. Pausata et al., 2012) have used satellite data only to evaluate models and not to examine if the model composition – NAO interactions can been seen in the satellite data itself. Finally, many studies (e.g. Pausata et al., 2012; and Thomas and Devasthale 2014) have looked at 1 or 2 pollutants in isolation. Our study analyses several species. Therefore, we feel that this is a timely study building upon previous work which uses a novel synergy of model and satellite datasets to investigate the influence of the NAO on multiple tropospheric (UTLS) pollutants (e.g. the anti-correlation in the $NO_x$ and ozone responses to the NAO).

We have updated the Introduction to explain this on P4 L10-20 "In this study, we aim to better constrain previously investigated relationships (e.g. NAO - $TCNO_2$, Eckhardt et al., 2003) and quantity unexplored relationships (e.g. vertical ozone profiles and upper troposphere – lower stratosphere (UTLS) peroxyacetyl nitrate (PAN)) between the NAO and atmospheric composition by utilising recent satellite observations with higher spatial resolution, more frequent sampling and smaller uncertainties (e.g. Ozone Monitoring Instrument (OMI) $TCNO_2$) and simulations from the TOMCAT chemistry transport model (CTM). These tools allow for a more comprehensive assessment of recent interactions between the NAO and tropospheric composition, the correlation response between trace gases, the extent to which the NAO can influence UTLS composition and an understanding of the key processes governing pollution levels over source regions. Section 2 discusses the observations and model setup, Section 3 describes the links between satellite observed/model composition and the NAO and our discussion and conclusions are presented in Sections 4 and 5.".

1.2. What is the justification for using these satellite datasets?

Eckhardt et al., (2003) investigated the impact of the NAO on satellite observed (GOME) $TCNO_2$ finding a reduction in $TCNO_2$ over the UK/north-western Europe. The enhancement of the westerly flow in NAO+ yielded a reduction in $TCNO_2$ linked to its long-range transport. Though a useful result from Eckhardt et al., (2003), the signal of the order of $10^{14}$ molecules/cm$^2$ is rather weak compared to the average concentrations. The differences are also the NAO+ composite minus the NAO- composite, while here we use the seasonal average to find the impact of the NAO on $TCNO_2$. Finally, given the large uncertainties in the DOAS technique, I would expect signals of this magnitude to likely be well within the uncertainty range of the composition averages in each grid cell over the UK/north-western Europe.

Therefore, our use of the OMI TCNO$_2$, which is a spatially higher resolution instrument with a larger sample size, finds similar signals over the UK/north-western Europe but outside the uncertainty of the instrument with anomalies in the order of $10^{15}$ molecules/cm$^2$ highlighting a more robust signal.

In term of the manuscript text, we have added the following text on P6 L11 "We primarily use OMI TCNO$_2$ data as the instrument has a much higher spatial resolution and sampling than the GOME TCNO$_2$ data used by Eckhardt et al., (2003). Therefore, we use OMI to build upon that work and explore whether we can detect a more robust TCNO$_2$-NAO signal, which is challenging given the short lifetime of NO$_2$.".

For tropospheric ozone, only a few studies have looked at the interaction between ozone and the NAO. Creilson et al., (2003) used the TOR product to look at this interaction over the southern Europe during the spring time NAO. Therefore, they were limited to the investigation of just the tropospheric column and looked at a different season to our study. Secondly, Pausata et al., (2012) only used the TOR data to evaluate their model, before having a detailed look at the modelled linked between the NAO and ozone. Therefore, there is a clear gap to use satellite tropospheric ozone data (i.e. from TES) which can be used to investigate the responses of ozone to the NAO, and can also provide important vertical information. As we show in our results from Section 3.1.3, there are clear differences between the two phases in ozone response to the NAO in Zones 1 & 2 on opposite sides of the North Atlantic. These patterns have not be found previously using satellite data.

We have made this point clearer in the Section 1 on P6 L17 "Previous studies of the NAO impacts on ozone have only used satellite tropospheric column data either directly or to evaluate model simulations. Therefore, the vertical ozone profiles retrieved by TES provide the opportunity to better understand the vertical response of ozone from NAO circulation patterns.".

For MIPAS PAN, as far as we are aware, this data has never been used before to look at potential impacts of the NAO on UTLS composition. As stated in the Introduction, Thomas and Devasthale (2014) used AIRS CO at 500 mb to look at the impact of the NAO on CO over Scandinavia, but limited research has been done using satellite data to see the response of pollutants to the NAO in the UTLS. PAN has a lifetime of several months once in the upper troposphere, so is a potentially good tracer for looking at polluted air masses. By using the MIPAS PAN data and the TOMCAT output in combination, we are able to explain these interactions (see sections 3.1.2 & 3.2.2). This is especially true for NAO-, where Figure 3d, 10d and 11d show that the satellite can detect enhanced PAN over Iceland in NAO- in the UTLS, which is simulated by the model and then shown to be due to the vertical transport of polluted tropospheric air masses from southern/central Europe into the UTLS over Iceland. Therefore, the use of MIPAS data is novel and, with the help of a model, highlights pathways of pollution transport into the UTLS.

We have made this point more clear on P7 L5 "Few studies have directly used satellite measurements of composition to investigate the influence of the NAO on UTLS trace gas distributions. This is the first study to use satellite retrieved UTLS PAN, which has a lifetime of several months in the UTLS (Singh et al., 1996), to investigate the impact of NAO tropospheric circulation patterns on UTLS composition.".

1.3. What is the added benefit of the TOMCAT model simulation?

The three satellite instruments, while providing useful information, do not have the same temporal and spatial coverage of a CTM. OMI TCNO$_2$ is an integrated column value and provides no vertical/surface information. The TES ozone data provides useful vertical information down into the boundary layer but has poor spatial coverage. Therefore, it is difficult to get a clear spatial pattern from the data alone. Hence, we collate the data into 4 regions and investigate changes in the vertical

profiles in response to the NAO. However, TOMCAT is able to provide detailed surface fields and vertical profiles across the entire domain. Finally, the MIPAS PAN in the UTLS provides limited information on exchanges of pollutants from the lower troposphere into the upper troposphere. For instance, even though MIPAS PAN shows enhanced PAN over the Icelandic region in the UTLS during NAO-, it is the model simulations (see Figure 11d) which identifies the source of this UTLS PAN enhancement from vertical assent of polluted tropospheric air.

To make this more clear in the manuscript we have added, on P8 L16, "Here, the TOMCAT simulations help to diagnose the key processes governing the satellite derived NAO-composition relationships by providing information (e.g. the surface) where the satellite instruments cannot detect trace gases and offers full spatial/temporal data coverage.".

1.4. The conclusions are too generic and need expanding.

We agree that the conclusions need to be more focussed. We have rewritten them as:

"This study has used state-of-the-art satellite data records of atmospheric trace gases to identify recent influences of the North Atlantic Oscillation (NAO) on tropospheric composition over the North Atlantic and Europe. We have used tropospheric column $NO_2$ ($TCNO_2$) measurements from the Ozone Monitoring Instrument (OMI), which provides higher resolution and sampling than past instruments, to detect clear and significant responses (i.e. reduction in UK $TCNO_2$ during NAO-high) from NAO circulation patterns, building upon the initial signal reported by Eckhardt et al. (2003). Vertical profiles of ozone from the Tropospheric Emissions Spectrometer (TES) allow a detailed assessment of satellite-observed lower tropospheric ozone sampled under the NAO phases. Robust, statistically significant signals are found on both sides of the North Atlantic as a result of changes in the westerly circulation during the two NAO phases. Finally, peroxyacetyl nitrate (PAN) observations in the upper troposphere-lower stratosphere (UTLS) from the Michelson Interferometer for Passive Atmospheric Sounding (MIPAS) are exploited, given the long lifetime of PAN (several months, Singh et al., 1996), for the first time to investigate vertical transport of polluted tropospheric air masses into the UTLS under different NAO conditions.

Our results, supported by simulations from the TOMCAT chemistry transport model (CTM), confirm that primary pollutant (i.e. $NO_2$) concentrations are reduced (enhanced) under NAO-high (NAO-low) conditions over Europe, heavily dependent on the strength of the westerly flow across the Atlantic. However, secondary pollutants, such as ozone, have anti-correlated patterns as maritime air masses (ozone-enriched air formed downwind from North American primary pollutant emissions) disperse polluted European air masses under NAO-high conditions, significantly increasing the background ozone levels. Under NAO-low conditions, the slackening of the North Atlantic westerly flow, allows for the accumulation of primary pollutants over Europe, where ozone concentrations are further decreased by ozone titration (loss through reaction with nitric oxide, NO). Different responses to that over Europe are observed and simulated by TES and TOMCAT over the western North Atlantic where enhanced westerly flow (NAO-high conditions) yields lower ozone concentrations over eastern North America through pollutant (both ozone and ozone precursors) long range transport towards Europe. However, the weakening of the westerly flow (NAO-low conditions) allows ozone to accumulate/form over the region. We also find that NAO circulation is important for UTLS composition as polluted air masses (e.g. with high PAN content) originating from Europe during NAO-low (accumulation of lower tropospheric pollution) can propagate to this altitude resulting in elevated PAN concentrations over Iceland/Southern Greenland. Model simulations show UTLS ozone spatial patterns over the North Atlantic are strongly anti-correlated to that of PAN, where the two trace gases act as flags for polluted tropospheric and clean stratospheric air in the UTLS.

Overall, the use of recent satellite data sets, not used in context of the NAO before, and a model simulation have quantified the recent influences of the NAO on tropospheric composition and co-variability between pollutants.".

**Direct Responses to Review 1's Comments:**

In this paper, the authors combine multiple satellite observations with CTM simulations to characterise the overall distribution of 3 chemical species, NO2, PAN and O3. Whilst the presentation of satellite observations and CTM simulations is clear, the overall aim of the study is too generic – ''to investigate the impact of the NAO circulation on tropospheric composition''. As a result, the main conclusion of the paper is also very general and does not, in its current form, constitute a new finding. I recommend that this paper undergoes significant major revisions.

General comments

1. Abstract and introduction: The structure of the introduction does not move on from what is known already, towards the issue that remains unknown and will be studied. As a result, it is not possible to determine the innovative aspect of this research. The authors need to state clearly which gap in the scientific knowledge this paper aims to fill and set out some specific research questions or hypothesis.

Please see response in general points 1.1 and 1.4.

2. Results: The authors have chosen to present the results in 2 sections. First the observations of tropospheric composition (section 3.1) and second the model simulations of tropospheric composition (section 3.2). Currently, use of the CTM simulations to 'diagnose the relationships seen in the satellite data' relies on the CTM composite wind fields. A similar result could be achieved using re-analysis wind fields. What is the additional benefit of using a complex CTM?

Please see response in general point 1.3.

3. The main conclusion is that the NAO is an 'important driver of winter-time atmospheric composition across Europe'. As the authors themselves state in the introduction, several previous studies have reached the same conclusion and therefore this is not a novel result. Please can the authors focus their research question and use the high quality datasets available to them to answer it.

Please see response in general point 1.1 and 1.4.

Specific comments

1. Page 2 line 16: 'altitude' is usually used as a vertical distance measurement but here it could be interpreted as a latitudinal distance. Can this be re-worded to make its use clearer?

Yes the term "altitude" in this case does refer to "a vertical distance measurement" and not "a latitudinal distance". However, I'm not sure why the reader would assume altitude would mean "a latitudinal distance". This text to us appears to be clear and we plan to leave it as it is.

2. Page 2 line 18: The authors refer to transport from the troposphere into the UTLS. As part of the troposphere is in the UTLS by definition, should troposphere be midtroposphere here?

We have replaced "troposphere" with "mid-troposphere".

3. Page 3 line 10: Here and elsewhere the authors refer to enhanced westerly circulation as 'storm tracks'. Of course, the low-level eddy driven westerlies can be used as a proxy for the storm track but it should be stated that this is the assumption here.

We believe the term "storm track" is suitable as we are being consistent with the literature (i.e. Hurrell et al., 1995 and Osborn, 2006) which uses this terminology frequently.

4. Page 3 line 12: 're-orientation' should be reorientation I think.

We have changed this.

5. Page 4 line 10: The authors here claim that they are building on previous work but it is not clear from the introduction specifically what research question(s) they are addressing (see also general comment 1).

Please see response in general points 1.1 and 1.4.

6. Page 4 line 14: What is the advantage of higher resolution and more frequent observations? How does this allow the authors to discover that was not known before?

As stated in general point 1.2, the signal seen in tropospheric column $NO_2$ by Eckhardt et al., (2003) is small relative to the absolute column averages and uncertainties on older instruments such as GOME, which will be larger given the coarseness of the instrument and infrequency of the observations (i.e. larger random errors). Here, in Figure 2 we reproduce a similar signal to Eckhardt et al., (2003) over parts of north-western Europe (mainly the UK) in NAO-, but with a more robust result as the deviation from the average under NAO+ conditions is of the same order of magnitude to that of the absolute concentrations (i.e. $x10^{15}$ molecules/cm$^2$) and outside the satellite uncertainties. Plus the WRT has also been used to show the results are statistically significant in several regions of the domain at the 95% confidence level. We have addressed this point in the manuscript with the text added from general point 1.2.

7. Page 4 line 24: What time period is used to normalise the pressure data? Also, why does the study focus on wintertime only? In some situations November data is included, but not in all, why isn't the same period used for all of the data?

The pressure fields between 2006 and 2015 have been normalised by the 2006-2015 winter-time (DJF) standard deviation. This is suggested on P5 L2 where the blue dotted lines are stated to represent the 1.0 and -1.0 standard deviations. However, to make this clearer, we have added on P4 L26:

"(normalised by the time-series stand deviation)" after "*time-series*" and removed "*normalised*" on P4 L25.

We focus on the winter-time NAO because this is the season when the influence of the NAO is strongest on North Atlantic circulation, as suggested by Hurrell et al., (1995) and Obsorn (2006). This is also stated in the Introduction on P3 L3-5. However, we agree the focus of this winter season could be made more clearly, despite the points above, so the title of the manuscript has been changed to "Influence of the winter-time North Atlantic Oscillation on European tropospheric composition".

We have now redone the analysis of all observations and model species for the winter-time NAO defined as November-December-January-February (NDJF). The text throughout the manuscript has been modified to reflect the slight changes in the spatial patterns of the relationships caused by the inclusion of November data.

8. Page 5 line 28: The 10km wind composites during different phases of the NAO are included in figure 1 but only briefly referred to in the text. Why were these figures included and how do they relate to the subsequent analysis?

The main purpose of Figure 1 was to highlight the impact of the wintertime NAO on circulation patterns over the North Atlantic region. The plots at 10 km were included to help analyse the MIPAS PAN responses to the NAO phases. The links between pressure anomalies / circulation patterns in the UTLS (Figure 1) are briefly discussed in section 3.1.2 with the satellite data, but the reader is then referred from P 9 L 24 to section 3.2.2 on P14 L22. Here the text states, P14 L20-24 "*PAN accumulation over Iceland and Southern Greenland (25 ppbv) correlates with the large UTLS pressure increase shown in Figure 1d. Figure 10d highlights the significant enhancement of PAN over Iceland/Southern Greenland with positive anomalies, relative to the winter-time aver-age, of 5-10 pptv.*" Therefore, we are linking the enhancement of UTLS pressure over Iceland in NAO- to the observed and modelled enhancement in UTLS PAN. Discussion of Figure 11d highlights the accumulation of PAN over the Iceland latitude band in the zonal UTLS NAO- PAN anomalies where vertical transport of PAN is from the polluted lower troposphere over Europe.

9. Page 6 section 2.2: This section details the satellite observations used in the study. However, as the authors have not motivated their decision to focus on PAN, NO2 and O3 in the introduction, it reads like a list of available data sources, rather than the necessary data needed to answer the research question posed. Why are these observations rather than others used in the paper?

Please see response in general point 1.2.

10. Page 7 section 2.3: This section appears to contain a lot of information that is not necessary to interpret the results or to reproduce the study. For example, the representation of many chemical species not used in this study are described. Please can this be re-written to focus on the information needed to support the results and conclusion of this specific study.

We have shortened the model description, section 2.3, to:

"TOMCAT is a three-dimensional (3-D) off-line chemistry transport model (CTM; e.g. Chipperfield (2006)). ECMWF ERA-Interim meteorological analyses are used to force the model winds, temperature, and humidity (Dee et al., 2011). The standard TOMCAT tropospheric chemistry version uses 82 advected tracers and 229 gas-phase reactions (Emmonset al., 2015), which includes the extended tropospheric chemistry (ExTC) scheme (Monks et al., 2017). TOMCAT also includes heterogeneous $N_2O_5$ hydrolysis using on-line size-resolved aerosol from the Global Model of Aerosol Processes (GLOMAP) model (Mann et al., 2010). The model anthropogenic emissions come from the Streets v1.2 inventory, which is a composite of several regional emissions inventories (Emmons et al.,2015). The MACCity inventory (Granier et al., 2011) is used for the natural emissions and biomass burning emissions come from the Global Fire Emissions Database (GFED) v3.1 inventory (Randerson et al., 2013).The model was initialised in December 2005, using a restart (initialisation) file from previous simulations, and run for 2006 to 2015 at the 2.8 ∘ × 2.8 ∘ spatial resolution (Monks et al., 2017)."

We still include some information about the number of reactions and provide the Monks et al., (2017) reference for the extended chemistry scheme as this will be important for ozone formation. The information of $N_2O_5$ is also retained as this will be an important sink of $NO_x$ and thus PAN formation.

11. Page 8 line 20: How many satellite overpasses are used to create figure 2. Is the noisy data a result of the sampling frequency? If the satellite data were averaged over larger spatial areas would this help with the signal to noise problem?

For both NAO phases, the number of satellite overpasses range from about 30-50 in the southern half of the domain, while typically 0-15 in the northern part of the domain. Over the UK, Scotland typically

has approximately 10-20 observations and in southern England it ranges between 20-40. Therefore, the northern part of the domain has few observations (<10) and the signal is noisy and unreliable. However, south of 60°N the observations increase in frequency where more sensible spatial patterns can be seen. The data were mapped onto the 0.05° x 0.05° grid using a pixel slicing methodology to retain the majority of the information as discussed by Pope et al., (accepted, ASL). This reference will be added to the manuscript once it is published. The purpose of using the higher resolution data set was to get away from the problem of Eckhardt et al., (2003) where the low resolution of the GOME data led to a weak signal under the NAO+ phase. Here, we seen a response in the same order of magnitude (i.e. $10^{15}$) as the seasonal average, which is statistically significant over parts of the UK.

In section 2.2 on P6 L11, we have added "The OMI $TCNO_2$ data is mapped onto a high resolution 0.05°x0.05° grid using the pixel slicing methodology of Pope et al., (accepted).". Then on P7 L11 we have added "For OMI $TCNO_2$ $n$ is typically greater than 20-30 observations per grid cell southwards of 60°N, while $n$ ranges between 0 and 10 observations between 60-70°N. Therefore, the $TCNO_2$ signal is less robust northwards of 60°N.".

12. Page 9 line 11: Do the authors have a physical mechanism to explain the lack of significant anomalies in fig 2d? The equivalent CTM simulation (fig 7d) shows significant positive anomalies across western Europe. What is the reason for this difference? Can the CTM be used to understand better the lack of relationship seen in the observations?

There are several potential reasons why this might be the case. Firstly, OMI has peak sensitivity to retrieving $NO_2$ in the mid-upper troposphere. Therefore, as the more stable conditions in NAO- are more likely to trap $NO_2$ lower down in the boundary layer where OMI has less sensitivity and might understate the absolute $TCNO_2$. Secondly, as the satellite represents clear sky situations, the model may have an average longer $NO_2$ lifetime as cloudy conditions may reduce the photochemical loss of $NO_2$. Therefore, we have added these potential limitations to our discussion in the manuscript. See P12 L10: "Potential reasons for model-satellite NAO- anomaly differences (Figure 2d and 6d) included: 1) As OMI has peak retrieval sensitivity in the mid-upper troposphere it potentially underestimates the full $TCNO_2$ under NAO- conditions when the more stable conditions trap $NO_2$ in the boundary layer, 2) The model $NO_2$ life time in the NAO- composite is potentially longer than the satellite equivalent as it represents all sky conditions, while the satellite composite represents clear sky conditions only (i.e. more photochemical loss of $NO_2$).".

13. Page 10 line 13 and page 14 line 16: How does the higher tropopause aid vertical transport of PAN into the UTLS? Is the UTLS a region that moves with the tropopause or is it defined to be a fixed altitude region?

Here we used the dynamical tropopause definition (i.e. +/-2 potential vorticity units) which is not fixed and will depend on vertical transport. If upwards vertical transport is enhanced, this will push the tropopause to a higher altitude and will more likely promote exchanges of tropospheric air masses into the UTLS (i.e. the air mass passes over the +/-2 PVU line).Therefore, in Figure 11c & d, one can see that the green line (zonally averaged tropopause across the Atlantic) is larger in NAO- when vertical transport in the mid-latitudes yields a higher tropopause and movement of PAN into the UTLS. This explains the PAN enhancements seen in Figures 3d and 10d. To make this clearer, we have reworded the text on P10 L13 to "The higher tropopause signifies enhanced vertical transport, which in this case, is the propagation of polluted air masses (i.e. large PAN content) from further down in the troposphere into the UTLS". Secondly, we have changed the text on P14 L16 to "Vertical transport will also have an important impact in NAO+, as signified by the higher MIPAS-derived tropopause height (see SM), with propagation of polluted air masses, from the lower troposphere, into the UTLS".

14. Page 14 line 29: Here the authors claim that the CTM PAN and satellite observed PAN have 'some similarities'. Comparison of figures 3c and 10c, and 4c and 11c, show very different spatial patterns. What is the reason for this? Does this mean that the CTM cannot be used to diagnose the relationships seen in the observations as they cannot reproduce the broad features observed during different phases of the NAO?

We agree with the reviewer that the TOMCAT-MIPAS pattern are more similar in the UTLS under NAO-conditions. In terms of the absolute PAN spatial distributions Figure 3a and 10a do have similar patterns. However, there are discrepancies in the anomalies between Figure 3c and 10c. We disagree that the picture is different between 4c and 11c, however. Above 300 hPa (i.e. no information below this for MIPAS, so cannot compare with TOMCAT) the zonal anomalies are consistent with enhanced PAN anomalies between 300-200 hPa over latitudes 60-90°N and negative anomalies above 150 hPa. The reverse dipole with altitude between 30-50°N remains consistent as well. In Figure 3c and 10c, TOMCAT does simulate positive PAN anomalies, as seen by MIPAS, between 35-45°N and in the top right of the domain. There are also observed/simulated negative anomalies between 45-65°N, but we are not sure what is driving the spatial (zonal) disparity between the centre points of the negative anomaly clusters in Figures 3c and 10c. However, these are acknowledged on P15 L1-5. Therefore, we believe TOMCAT is a suitable model to diagnose the PAN response to the NAO (especially in NAO-). We have added the following text on P15 L5: "Therefore, the model results only allow for limited assessment of the NAO influence of UTLS PAN in NAO+ over these regions".

15. Page 20 line 8: Please remove 'successfully' from this sentence. It is up to the reader, not the authors to judge the success of the paper.

Done. We have removed "successfully".

**References:**

[revised manuscript text omitted]

---

## Author Comment (AC2) · 5 May 2018

**Response to Reviewer Comments on Manuscript "Influence of the North Atlantic Oscillation on European tropospheric composition: an observational and modelling study"**

We thank the reviewers for their comments and suggestions for the manuscript. The reviewer comments are given below in black text, followed by our responses in red text and any additions to the manuscript in blue text. References to page and line numbers are based on the ACPD version of the manuscript.

**Firstly, as all three reviewers had similar "general comments", we have prepared responses to these points applicable to all the reviewers. These are:**

1.1. Why is the work novel?

Several studies have indeed looked at the interaction of the NAO and composition using datasets (e.g. satellite or surface) and models. However, many of these studies were performed 10-15 years ago and new satellite missions (see point 1.2) allow us to build upon previous work (e.g. Eckhardt et al., 2003; Creilson et al., 2003) to see if these relationships seen in the past are still valid and if newer more sophisticated instruments see similar or different patterns. Secondly, some studies (e.g. Pausata et al., 2012) have used satellite data only to evaluate models and not to examine if the model composition – NAO interactions can been seen in the satellite data itself. Finally, many studies (e.g. Pausata et al., 2012; and Thomas and Devasthale 2014) have looked at 1 or 2 pollutants in isolation. Our study analyses several species. Therefore, we feel that this is a timely study building upon previous work which uses a novel synergy of model and satellite datasets to investigate the influence of the NAO on multiple tropospheric (UTLS) pollutants (e.g. the anti-correlation in the $NO_x$ and ozone responses to the NAO).

We have updated the Introduction to explain this on P4 L10-20 "In this study, we aim to better constrain previously investigated relationships (e.g. NAO - $TCNO_2$, Eckhardt et al., 2003) and quantity unexplored relationships (e.g. vertical ozone profiles and upper troposphere – lower stratosphere (UTLS) peroxyacetyl nitrate (PAN)) between the NAO and atmospheric composition by utilising recent satellite observations with higher spatial resolution, more frequent sampling and smaller uncertainties (e.g. Ozone Monitoring Instrument (OMI) $TCNO_2$) and simulations from the TOMCAT chemistry transport model (CTM). These tools allow for a more comprehensive assessment of recent interactions between the NAO and tropospheric composition, the correlation response between trace gases, the extent to which the NAO can influence UTLS composition and an understanding of the key processes governing pollution levels over source regions. Section 2 discusses the observations and model setup, Section 3 describes the links between satellite observed/model composition and the NAO and our discussion and conclusions are presented in Sections 4 and 5.".

1.2. What is the justification for using these satellite datasets?

Eckhardt et al., (2003) investigated the impact of the NAO on satellite observed (GOME) $TCNO_2$ finding a reduction in $TCNO_2$ over the UK/north-western Europe. The enhancement of the westerly flow in NAO+ yielded a reduction in $TCNO_2$ linked to its long-range transport. Though a useful result from Eckhardt et al., (2003), the signal of the order of $10^{14}$ molecules/cm$^2$ is rather weak compared to the average concentrations. The differences are also the NAO+ composite minus the NAO- composite, while here we use the seasonal average to find the impact of the NAO on $TCNO_2$. Finally, given the large uncertainties in the DOAS technique, I would expect signals of this magnitude to likely be well within the uncertainty range of the composition averages in each grid cell over the UK/north-western Europe.

Therefore, our use of the OMI TCNO$_2$, which is a spatially higher resolution instrument with a larger sample size, finds similar signals over the UK/north-western Europe but outside the uncertainty of the instrument with anomalies in the order of 10$^{15}$ molecules/cm$^2$ highlighting a more robust signal.

In term of the manuscript text, we have added the following text on P6 L11 "We primarily use OMI TCNO$_2$ data as the instrument has a much higher spatial resolution and sampling than the GOME TCNO$_2$ data used by Eckhardt et al., (2003). Therefore, we use OMI to build upon that work and explore whether we can detect a more robust TCNO$_2$-NAO signal, which is challenging given the short lifetime of NO$_2$.".

For tropospheric ozone, only a few studies have looked at the interaction between ozone and the NAO. Creilson et al., (2003) used the TOR product to look at this interaction over the southern Europe during the spring time NAO. Therefore, they were limited to the investigation of just the tropospheric column and looked at a different season to our study. Secondly, Pausata et al., (2012) only used the TOR data to evaluate their model, before having a detailed look at the modelled linked between the NAO and ozone. Therefore, there is a clear gap to use satellite tropospheric ozone data (i.e. from TES) which can be used to investigate the responses of ozone to the NAO, and can also provide important vertical information. As we show in our results from Section 3.1.3, there are clear differences between the two phases in ozone response to the NAO in Zones 1 & 2 on opposite sides of the North Atlantic. These patterns have not be found previously using satellite data.

We have made this point clearer in the Section 1 on P6 L17 "Previous studies of the NAO impacts on ozone have only used satellite tropospheric column data either directly or to evaluate model simulations. Therefore, the vertical ozone profiles retrieved by TES provide the opportunity to better understand the vertical response of ozone from NAO circulation patterns.".

For MIPAS PAN, as far as we are aware, this data has never been used before to look at potential impacts of the NAO on UTLS composition. As stated in the Introduction, Thomas and Devasthale (2014) used AIRS CO at 500 mb to look at the impact of the NAO on CO over Scandinavia, but limited research has been done using satellite data to see the response of pollutants to the NAO in the UTLS. PAN has a lifetime of several months once in the upper troposphere, so is a potentially good tracer for looking at polluted air masses. By using the MIPAS PAN data and the TOMCAT output in combination, we are able to explain these interactions (see sections 3.1.2 & 3.2.2). This is especially true for NAO-, where Figure 3d, 10d and 11d show that the satellite can detect enhanced PAN over Iceland in NAO- in the UTLS, which is simulated by the model and then shown to be due to the vertical transport of polluted tropospheric air masses from southern/central Europe into the UTLS over Iceland. Therefore, the use of MIPAS data is novel and, with the help of a model, highlights pathways of pollution transport into the UTLS.

We have made this point more clear on P7 L5 "Few studies have directly used satellite measurements of composition to investigate the influence of the NAO on UTLS trace gas distributions. This is the first study to use satellite retrieved UTLS PAN, which has a lifetime of several months in the UTLS (Singh et al., 1996), to investigate the impact of NAO tropospheric circulation patterns on UTLS composition.".

1.3. What is the added benefit of the TOMCAT model simulation?

The three satellite instruments, while providing useful information, do not have the same temporal and spatial coverage of a CTM. OMI TCNO$_2$ is an integrated column value and provides no vertical/surface information. The TES ozone data provides useful vertical information down into the boundary layer but has poor spatial coverage. Therefore, it is difficult to get a clear spatial pattern from the data alone. Hence, we collate the data into 4 regions and investigate changes in the vertical

profiles in response to the NAO. However, TOMCAT is able to provide detailed surface fields and vertical profiles across the entire domain. Finally, the MIPAS PAN in the UTLS provides limited information on exchanges of pollutants from the lower troposphere into the upper troposphere. For instance, even though MIPAS PAN shows enhanced PAN over the Icelandic region in the UTLS during NAO-, it is the model simulations (see Figure 11d) which identifies the source of this UTLS PAN enhancement from vertical assent of polluted tropospheric air.

To make this more clear in the manuscript we have added, on P8 L16, "Here, the TOMCAT simulations help to diagnose the key processes governing the satellite derived NAO-composition relationships by providing information (e.g. the surface) where the satellite instruments cannot detect trace gases and offers full spatial/temporal data coverage.".

1.4. The conclusions are too generic and need expanding.

We agree that the conclusions need to be more focussed. We have rewritten them as:

"This study has used state-of-the-art satellite data records of atmospheric trace gases to identify recent influences of the North Atlantic Oscillation (NAO) on tropospheric composition over the North Atlantic and Europe. We have used tropospheric column $NO_2$ (TCNO$_2$) measurements from the Ozone Monitoring Instrument (OMI), which provides higher resolution and sampling than past instruments, to detect clear and significant responses (i.e. reduction in UK TCNO$_2$ during NAO-high) from NAO circulation patterns, building upon the initial signal reported by Eckhardt et al. (2003). Vertical profiles of ozone from the Tropospheric Emissions Spectrometer (TES) allow a detailed assessment of satellite-observed lower tropospheric ozone sampled under the NAO phases. Robust, statistically significant signals are found on both sides of the North Atlantic as a result of changes in the westerly circulation during the two NAO phases. Finally, peroxyacetyl nitrate (PAN) observations in the upper troposphere-lower stratosphere (UTLS) from the Michelson Interferometer for Passive Atmospheric Sounding (MIPAS) are exploited, given the long lifetime of PAN (several months, Singh et al., 1996), for the first time to investigate vertical transport of polluted tropospheric air masses into the UTLS under different NAO conditions.

Our results, supported by simulations from the TOMCAT chemistry transport model (CTM), confirm that primary pollutant (i.e. $NO_2$) concentrations are reduced (enhanced) under NAO-high (NAO-low) conditions over Europe, heavily dependent on the strength of the westerly flow across the Atlantic. However, secondary pollutants, such as ozone, have anti-correlated patterns as maritime air masses (ozone-enriched air formed downwind from North American primary pollutant emissions) disperse polluted European air masses under NAO-high conditions, significantly increasing the background ozone levels. Under NAO-low conditions, the slackening of the North Atlantic westerly flow, allows for the accumulation of primary pollutants over Europe, where ozone concentrations are further decreased by ozone titration (loss through reaction with nitric oxide, NO). Different responses to that over Europe are observed and simulated by TES and TOMCAT over the western North Atlantic where enhanced westerly flow (NAO-high conditions) yields lower ozone concentrations over eastern North America through pollutant (both ozone and ozone precursors) long range transport towards Europe. However, the weakening of the westerly flow (NAO-low conditions) allows ozone to accumulate/form over the region. We also find that NAO circulation is important for UTLS composition as polluted air masses (e.g. with high PAN content) originating from Europe during NAO-low (accumulation of lower tropospheric pollution) can propagate to this altitude resulting in elevated PAN concentrations over Iceland/Southern Greenland. Model simulations show UTLS ozone spatial patterns over the North Atlantic are strongly anti-correlated to that of PAN, where the two trace gases act as flags for polluted tropospheric and clean stratospheric air in the UTLS.

Overall, the use of recent satellite data sets, not used in context of the NAO before, and a model simulation have quantified the recent influences of the NAO on tropospheric composition and co-variability between pollutants.".

**Direct Responses to Review 2's Comments:**

The authors used satellite observations and the TOMCAT chemistry transport model (CTM) to investigate the influence of the winter-time North Atlantic Oscillation (NAO) phases on the tropospheric concentrations of NO2, PAN, ozone over the North Atlantic and western Europe. Though the methods are scientifically sound, the authors fail to properly motivate the study and it is not clear what is the underlying purpose other than to ascertain what was presented in numerous other studies that the manuscript cites.

Please see the response to general point 1.1.

The findings and conclusions of the manuscript in my view do not offer any new general implications for atmospheric science. The manuscript fails to expand beyond what is readily established in the literature or introduce sufficiently novel methods or techniques, other than perhaps some incremental improvements.

Please see the response to general point 1.4.

My recommendation is that this paper undergoes major revisions before publication in ACP to: properly motivate the choice of remote sensing products, explain the need and purpose of supplementing the analysis of observations with model simulations, and to include more detailed discussion of the scientific implications of the outcomes.

Please see the response to general points 1.1 to 1.4.

Specific comments:

p.4 l.24: It's understood that November is sometimes included in the seasonal data to increase low statistics. However, to aid intercomparison wouldn't it be better to be consistent in all cases?

We have now redone the analysis of all observations and model species for the winter-time NAO defined as November-December-January-February (NDJF). The text throughout the manuscript has been modified to reflect the slight changes in the spatial patterns of the relationships caused by the inclusion of November data.

p.5 l.1: "significant" NAO+ and NAO- phases may be misconstrued by the reader. Propose to change to "high and low" as is usually found in the literature.

The reviewer is correct that "high" and "low" is often used to represent significant NAO+ and NAO- events in the literature. To that end, we have replace occurrences of "NAO+" and "NAO-" with "NAO-high" and "NAO-low", respectively. This introduction of NAO+/- in the abstract and section 2.1 (NAOI) has also been re-defined.

p.5. l.28: What is the point of including wind vectors in the Figs? They are not referred to in the discussion/analysis.

The wind vectors allow the reader to see the circulation patterns in each NAO phase and the impact on composition. The analysis of the circulation patterns are discussed in section 2.1 relating to Figure 1, in section 3.2.2 discussing the PAN spatial distribution in Figure 10, in section 3.2.3 (e.g. *P15 L1-2 "TOMCAT surface ozone under NAO+ (Figure 12a) peaks at approximately 28-30 ppbv over sub-tropical*

*and western North Atlantic co-located with the enhanced westerlies.*" and several other locations. Therefore, we need to leave the wind vectors on the Figures in their current state.

p.7.l.7: Please clarify what is meant by "The systematic errors will cancel considerably when comparing species-NAO composites to their winter-time averages". Why is that so?

This text is trying to point out that the systemic errors (offsets), absolute or percentage, are consistent between measurements i.e. unlike random errors which will vary considerably for each observation. Therefore, when the seasonal composition average is subtracted from the NAO composite, any systematic errors will largely cancel leaving the residual NAO anomaly and associated random errors. Therefore, to make this clearer, the text has been reworded to: "When each chemical species is sampled under the NAO phases and then compared with the seasonal (winter-time) average, the anomalies (i.e. NAO composite – seasonal average) will be dominated by random errors as the systematic errors will cancel considerably".

p7.l.14: "range from approximately 10-40%, peak at 15-20% and are between 10-20%" meaning needs to be made clearer. Is that respectively for OMI/MIPAS/TES?

Yes this is for the respective instruments. We have reworded P7 L12-15 with "However, over the North Atlantic and western Europe, OMI $TCNO_2$ random errors range between approximately 10-40%. For MIPAS PAN at 150 hPa the random errors peak at 15-20%, while ranging between 10-20% for TES lower tropospheric ozone."

Sec. 2.3: The section on the TOMCAT model has to be reworked to limit to information that is relevant to the present study.

We have shortened the model description, section 2.3, to:

"TOMCAT is a three-dimensional (3-D) off-line chemistry transport model (CTM; e.g. Chipperfield (2006)). ECMWF ERA-Interim meteorological analyses are used to force the model winds, temperature, and humidity (Dee et al., 2011). The standard TOMCAT tropospheric chemistry version uses 82 advected tracers and 229 gas-phase reactions (Emmonset al., 2015), which includes the extended tropospheric chemistry (ExTC) scheme (Monks et al., 2017). TOMCAT also includes heterogeneous $N_2O_5$ hydrolysis using on-line size-resolved aerosol from the Global Model of Aerosol Processes (GLOMAP) model (Mann et al., 2010). The model anthropogenic emissions come from the Streets v1.2 inventory, which is a composite of several regional emissions inventories (Emmons et al.,2015). The MACCity inventory (Granier et al., 2011) is used for the natural emissions and biomass burning emissions come from the Global Fire Emissions Database (GFED) v3.1 inventory (Randerson et al., 2013).The model was initialised in December 2005, using a restart (initialisation) file from previous simulations, and run for 2006 to 2015 at the 2.8 ◦ × 2.8 ◦ spatial resolution (Monks et al., 2017)."

We still include some information about the number of reactions and provide the Monks et al., (2017) reference for the extended chemistry scheme as this will be important for ozone formation. The information of $N_2O_5$ is also retained as this will be an important sink of $NO_x$ and thus PAN formation.

p.10 l.12: Any reference as to why the higher tropopause potentially aids vertical transport into the UTLS?

Here we used the dynamical tropopause definition (i.e. +/-2 potential vorticity units) which is not fixed and will depend on vertical transport. If upwards vertical transport is enhanced, this will push the tropopause to a higher altitude and will more likely promote exchanges of tropospheric air masses

into the UTLS (i.e. the air mass passes over the +/-2 PVU line).Therefore, in Figure 11c & d, one can see that the green line (zonally averaged tropopause across the Atlantic) is larger in NAO- when vertical transport in the mid-latitudes yields a higher tropopause and movement of PAN into the UTLS. This explains the PAN enhancements seen in Figures 3d and 10d. To make this clearer, we have reworded the text on P10 L13 to "The higher tropopause signifies enhanced vertical transport, which in this case, is the propagation of polluted air masses (i.e. large PAN content) from further down in the troposphere into the UTLS". Secondly, we have changed the text on P14 L16 to "Vertical transport will also have an important impact in NAO+, as signified by the higher MIPAS-derived tropopause height (see SM), with propagation of polluted air masses, from the lower troposphere, into the UTLS".

p.11 l.21: "We also have evaluated TOMCAT surface/tropospheric ozone against a range of observations. In all cases, TOMCAT has suitable skill to represent these chemical tracers and their responses to the NAO circulation patterns.". Is there any appropriate reference? OR Perhaps include the comparison in the SM?

This is all discussed or done in the supplementary material. Hence we have added "(see SM)" on P11 L24.

The NO2 has a much short lifetime in the atmosphere than the timespan considered here and is heavily dependent on emissions. It is not convincing that the randomly scattered green regions of significance in Fig. 2 allow or support any generalised conclusions for NO2 concentrations to be influenced by a seasonal teleconnection pattern.

The reviewer is correct that the NO2 signal is not as strong as the TES ozone or MIPAS PAN signals, however, we believe the results in Figure 2 do provide some insight into the impact of the NAO on NO2. Eckhardt et al., (2003) found a weak response of $TCNO_2$ under NAO+, which we have built upon highlighting a significant reduction over the UK (Figure 2c). This is seen to a degree between Figure 2a and 2b where the $TCNO_2$ concentrations over the UK in NAO+ range from 5-10 x$10^{15}$molecules/cm$^2$ while 8-12 x$10^{15}$molecules/cm$^2$ during NAO-. These results potentially lay the foundations to extend this work once the new high resolution TropOMI instrument (7x7 km$^2$ vs 10s x 10s km$^2$ for OMI) $TCNO_2$ product becomes available and city scale changes potentially can be seen. Please see the response to general point 4 and additional text to the manuscript where we emphasise these results.

Technical corrections:

p.6 l.3: change to "measurements of total column NO2 (TCNO2)"

This should not be necessary as TCNO2 has already been defined on P3 L17.

p.6 l.4: UTLS PAN -> "UTLS Peroxyacetyl Nitrate (PAN)"

This has been changed in line with the reviewer's comment.

p.7 l.1 Printing the full file type list in the manuscript text is unecessary and extraneous. Either refer to it (or move details in a supplement).

We agree and this information has been placed in the supplementary material.

p.7 l.6 and elsewhere "random error" -> "statistical uncertainty"

We believe that "random error" is a suitable term to use as we provide discussion on the different type of errors (i.e. systemic and random). Therefore, it is better to be consistent.

p.8 l.8: Cite IUPAC website as reference

In line with Reviewer 1's Specific Comment 10, the model description text has been shortened and the reference to IUPAC removed.

Fig.3: It's very hard for the reader to discern the green lines and areas of significance on top of dashed lines in panels c) d). Perhaps make bolder?

We have made the green lines thicker on figures with the anomaly plots and significance contouring.

Sec. 3.2 Title "Model Composition" -> "Model Results"

This has been changed.

p.12 l.4,6: Missing TCNO2 units

This has been changed.

p.14 l.14: larger -> higher

This has been changed.

Fig.5 Squares and diamonds are cluttered over the errors. Perhaps a different representation using horizontal bands and removal of the yellow sensitivity band may be clearer.

We feel the current graphic is suitable and is the same presentation methodology of Pope et al., (2016). The copy editors at ACP will be able to take the decision on clarity of the final figures.

[revised manuscript text omitted]

---

## Author Comment (AC3) · 5 May 2018

**Response to Reviewer Comments on Manuscript "Influence of the North Atlantic Oscillation on European tropospheric composition: an observational and modelling study"**

We thank the reviewers for their comments and suggestions for the manuscript. The reviewer comments are given below in black text, followed by our responses in red text and any additions to the manuscript in blue text. References to page and line numbers are based on the ACPD version of the manuscript.

**Firstly, as all three reviewers had similar "general comments", we have prepared responses to these points applicable to all the reviewers. These are:**

1.1. Why is the work novel?

Several studies have indeed looked at the interaction of the NAO and composition using datasets (e.g. satellite or surface) and models. However, many of these studies were performed 10-15 years ago and new satellite missions (see point 1.2) allow us to build upon previous work (e.g. Eckhardt et al., 2003; Creilson et al., 2003) to see if these relationships seen in the past are still valid and if newer more sophisticated instruments see similar or different patterns. Secondly, some studies (e.g. Pausata et al., 2012) have used satellite data only to evaluate models and not to examine if the model composition – NAO interactions can been seen in the satellite data itself. Finally, many studies (e.g. Pausata et al., 2012; and Thomas and Devasthale 2014) have looked at 1 or 2 pollutants in isolation. Our study analyses several species. Therefore, we feel that this is a timely study building upon previous work which uses a novel synergy of model and satellite datasets to investigate the influence of the NAO on multiple tropospheric (UTLS) pollutants (e.g. the anti-correlation in the $NO_x$ and ozone responses to the NAO).

We have updated the Introduction to explain this on P4 L10-20 "In this study, we aim to better constrain previously investigated relationships (e.g. NAO - $TCNO_2$, Eckhardt et al., 2003) and quantity unexplored relationships (e.g. vertical ozone profiles and upper troposphere – lower stratosphere (UTLS) peroxyacetyl nitrate (PAN)) between the NAO and atmospheric composition by utilising recent satellite observations with higher spatial resolution, more frequent sampling and smaller uncertainties (e.g. Ozone Monitoring Instrument (OMI) $TCNO_2$) and simulations from the TOMCAT chemistry transport model (CTM). These tools allow for a more comprehensive assessment of recent interactions between the NAO and tropospheric composition, the correlation response between trace gases, the extent to which the NAO can influence UTLS composition and an understanding of the key processes governing pollution levels over source regions. Section 2 discusses the observations and model setup, Section 3 describes the links between satellite observed/model composition and the NAO and our discussion and conclusions are presented in Sections 4 and 5.".

1.2. What is the justification for using these satellite datasets?

Eckhardt et al., (2003) investigated the impact of the NAO on satellite observed (GOME) $TCNO_2$ finding a reduction in $TCNO_2$ over the UK/north-western Europe. The enhancement of the westerly flow in NAO+ yielded a reduction in $TCNO_2$ linked to its long-range transport. Though a useful result from Eckhardt et al., (2003), the signal of the order of $10^{14}$ molecules/cm$^2$ is rather weak compared to the average concentrations. The differences are also the NAO+ composite minus the NAO- composite, while here we use the seasonal average to find the impact of the NAO on $TCNO_2$. Finally, given the large uncertainties in the DOAS technique, I would expect signals of this magnitude to likely be well within the uncertainty range of the composition averages in each grid cell over the UK/north-western Europe.

Therefore, our use of the OMI TCNO$_2$, which is a spatially higher resolution instrument with a larger sample size, finds similar signals over the UK/north-western Europe but outside the uncertainty of the instrument with anomalies in the order of $10^{15}$ molecules/cm$^2$ highlighting a more robust signal.

In term of the manuscript text, we have added the following text on P6 L11 "We primarily use OMI TCNO$_2$ data as the instrument has a much higher spatial resolution and sampling than the GOME TCNO$_2$ data used by Eckhardt et al., (2003). Therefore, we use OMI to build upon that work and explore whether we can detect a more robust TCNO$_2$-NAO signal, which is challenging given the short lifetime of NO$_2$.".

For tropospheric ozone, only a few studies have looked at the interaction between ozone and the NAO. Creilson et al., (2003) used the TOR product to look at this interaction over the southern Europe during the spring time NAO. Therefore, they were limited to the investigation of just the tropospheric column and looked at a different season to our study. Secondly, Pausata et al., (2012) only used the TOR data to evaluate their model, before having a detailed look at the modelled linked between the NAO and ozone. Therefore, there is a clear gap to use satellite tropospheric ozone data (i.e. from TES) which can be used to investigate the responses of ozone to the NAO, and can also provide important vertical information. As we show in our results from Section 3.1.3, there are clear differences between the two phases in ozone response to the NAO in Zones 1 & 2 on opposite sides of the North Atlantic. These patterns have not be found previously using satellite data.

We have made this point clearer in the Section 1 on P6 L17 "Previous studies of the NAO impacts on ozone have only used satellite tropospheric column data either directly or to evaluate model simulations. Therefore, the vertical ozone profiles retrieved by TES provide the opportunity to better understand the vertical response of ozone from NAO circulation patterns.".

For MIPAS PAN, as far as we are aware, this data has never been used before to look at potential impacts of the NAO on UTLS composition. As stated in the Introduction, Thomas and Devasthale (2014) used AIRS CO at 500 mb to look at the impact of the NAO on CO over Scandinavia, but limited research has been done using satellite data to see the response of pollutants to the NAO in the UTLS. PAN has a lifetime of several months once in the upper troposphere, so is a potentially good tracer for looking at polluted air masses. By using the MIPAS PAN data and the TOMCAT output in combination, we are able to explain these interactions (see sections 3.1.2 & 3.2.2). This is especially true for NAO-, where Figure 3d, 10d and 11d show that the satellite can detect enhanced PAN over Iceland in NAO- in the UTLS, which is simulated by the model and then shown to be due to the vertical transport of polluted tropospheric air masses from southern/central Europe into the UTLS over Iceland. Therefore, the use of MIPAS data is novel and, with the help of a model, highlights pathways of pollution transport into the UTLS.

We have made this point more clear on P7 L5 "Few studies have directly used satellite measurements of composition to investigate the influence of the NAO on UTLS trace gas distributions. This is the first study to use satellite retrieved UTLS PAN, which has a lifetime of several months in the UTLS (Singh et al., 1996), to investigate the impact of NAO tropospheric circulation patterns on UTLS composition.".

1.3. What is the added benefit of the TOMCAT model simulation?

The three satellite instruments, while providing useful information, do not have the same temporal and spatial coverage of a CTM. OMI TCNO$_2$ is an integrated column value and provides no vertical/surface information. The TES ozone data provides useful vertical information down into the boundary layer but has poor spatial coverage. Therefore, it is difficult to get a clear spatial pattern from the data alone. Hence, we collate the data into 4 regions and investigate changes in the vertical

profiles in response to the NAO. However, TOMCAT is able to provide detailed surface fields and vertical profiles across the entire domain. Finally, the MIPAS PAN in the UTLS provides limited information on exchanges of pollutants from the lower troposphere into the upper troposphere. For instance, even though MIPAS PAN shows enhanced PAN over the Icelandic region in the UTLS during NAO-, it is the model simulations (see Figure 11d) which identifies the source of this UTLS PAN enhancement from vertical assent of polluted tropospheric air.

To make this more clear in the manuscript we have added, on P8 L16, "Here, the TOMCAT simulations help to diagnose the key processes governing the satellite derived NAO-composition relationships by providing information (e.g. the surface) where the satellite instruments cannot detect trace gases and offers full spatial/temporal data coverage.".

1.4. The conclusions are too generic and need expanding.

We agree that the conclusions need to be more focussed. We have rewritten them as:

"This study has used state-of-the-art satellite data records of atmospheric trace gases to identify recent influences of the North Atlantic Oscillation (NAO) on tropospheric composition over the North Atlantic and Europe. We have used tropospheric column $NO_2$ ($TCNO_2$) measurements from the Ozone Monitoring Instrument (OMI), which provides higher resolution and sampling than past instruments, to detect clear and significant responses (i.e. reduction in UK $TCNO_2$ during NAO-high) from NAO circulation patterns, building upon the initial signal reported by Eckhardt et al. (2003). Vertical profiles of ozone from the Tropospheric Emissions Spectrometer (TES) allow a detailed assessment of satellite-observed lower tropospheric ozone sampled under the NAO phases. Robust, statistically significant signals are found on both sides of the North Atlantic as a result of changes in the westerly circulation during the two NAO phases. Finally, peroxyacetyl nitrate (PAN) observations in the upper troposphere-lower stratosphere (UTLS) from the Michelson Interferometer for Passive Atmospheric Sounding (MIPAS) are exploited, given the long lifetime of PAN (several months, Singh et al., 1996), for the first time to investigate vertical transport of polluted tropospheric air masses into the UTLS under different NAO conditions.

Our results, supported by simulations from the TOMCAT chemistry transport model (CTM), confirm that primary pollutant (i.e. $NO_2$) concentrations are reduced (enhanced) under NAO-high (NAO-low) conditions over Europe, heavily dependent on the strength of the westerly flow across the Atlantic. However, secondary pollutants, such as ozone, have anti-correlated patterns as maritime air masses (ozone-enriched air formed downwind from North American primary pollutant emissions) disperse polluted European air masses under NAO-high conditions, significantly increasing the background ozone levels. Under NAO-low conditions, the slackening of the North Atlantic westerly flow, allows for the accumulation of primary pollutants over Europe, where ozone concentrations are further decreased by ozone titration (loss through reaction with nitric oxide, NO). Different responses to that over Europe are observed and simulated by TES and TOMCAT over the western North Atlantic where enhanced westerly flow (NAO-high conditions) yields lower ozone concentrations over eastern North America through pollutant (both ozone and ozone precursors) long range transport towards Europe. However, the weakening of the westerly flow (NAO-low conditions) allows ozone to accumulate/form over the region. We also find that NAO circulation is important for UTLS composition as polluted air masses (e.g. with high PAN content) originating from Europe during NAO-low (accumulation of lower tropospheric pollution) can propagate to this altitude resulting in elevated PAN concentrations over Iceland/Southern Greenland. Model simulations show UTLS ozone spatial patterns over the North Atlantic are strongly anti-correlated to that of PAN, where the two trace gases act as flags for polluted tropospheric and clean stratospheric air in the UTLS.

Overall, the use of recent satellite data sets, not used in context of the NAO before, and a model simulation have quantified the recent influences of the NAO on tropospheric composition and co-variability between pollutants.".

**Direct Responses to Review 3's Comments:**

I was going to write a detailed review, but, I noticed that the other two reviewers have already broadly pointed out the concerns I had when I was reading the manuscript. In my opinion, the focus of the study is misplaced and the message is lost in trying to demonstrate/show different things instead of focussing on one particular topic. For example, it is known since few decades that NAO has a strong influence on northern European pollution variability. However, that does not rule out that one should not study this any more as our understanding of the processes and the tools constantly improves.

So, I will not criticize the main motivation behind this study, however, I do feel that the authors should have focussed on either revealing/discussing a new mechansim or complementing the existing ones. The way the study has concluded is too vague to delineate precisely what new knowledge has been gained. Maybe, the authors should take a step back and re-think what their results really convey. I also, have a few other points that the authors might consider improving.

Please see the response to general points 1.1 and 1.4.

1. I do not understand why the authors chose these three species, TCNO2, O3 and PAN and what is their interdependency and why should we study their covariability.

Please see the response to general point 1.2.

2. The selection of the regions is not properly motivated (Fig.5).

TES has poor spatial coverage so it is difficult to get robust pollutant maps. Therefore, we have clustered the individual profiless into four domains to get vertical profiles for each NAO phase. The domains were chosen cover the Atlantic to try and see how the ozone responded to the NAO in different regions. To make this clearer, we have added on P11 L1 "These four domains are selected because TES has infrequent spatial sampling (Richards et al., 2008) meaning spatial ozone distributions are often noisy/unclear.".

3. It is not clear what have we actually learned from using the model in addition to observations.

Please see the response to general point 1.3.

**References:**

[revised manuscript text omitted]